# FINESPLAT: FINE-GRAINED 3D OPEN-VOCABULARY LANGUAGE GAUSSIAN SPLATTING

## ABSTRACT

Existing open-vocabulary scene understanding methods are primarily limited to coarse-grained understanding at the object category level, making them incapable of handling fine-grained queries. In this paper, we introduce a challenging task of fine-grained open-vocabulary scene understanding and propose a novel fine-grained 3D language gaussian splatting framework, FineSplat for short. Unlike prior methods that rely on the vision-language alignment model, such as CLIP, FineSplat models the feature field solely from textual captions, transforming the cross-modal feature matching challenge into a retrieval process between queries and captions. Specifically, we design the Fine-Grained Caption Generation (FGCG) strategy to obtain captions containing multi-dimensional fine-grained attributes. Then, the Fine-Grained Feature Field Modeling (FGFFM) strategy is introduced to encode generated fine-grained captions into object-level semantic features, which subsequently supervise the training of 3D Gaussian representations. Furthermore, we construct Fine-OVS, a benchmark to support research and evaluation of the fine-grained open-vocabulary scene understanding task. Extensive experiments conducted on the Fine-OVS demonstrate that our FineSplat framework significantly outperforms existing state-of-the-art methods.

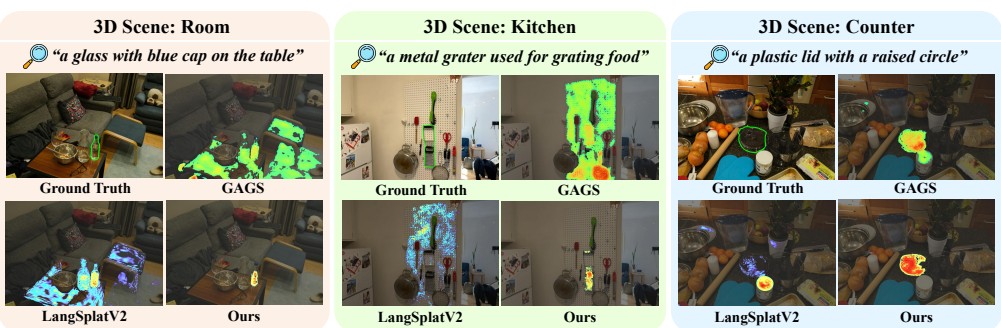

Figure 1: Visualization results of FineSplat and previous state-of-the-art (SOTA) methods on fine-grained open-vocabulary 3D scene understanding. When given queries with multi-dimensional fine-grained attributes, GAGS (Peng et al., 2025) and LangSplatV2 (Li et al., 2025) often obtain erroneous results due to feature mismatches, whereas our proposed method accurately activates the target objects, enabling effective fine-grained understanding of 3D scenes.

## 1 INTRODUCTION

Open-vocabulary 3D scene understanding has emerged as a key challenge in computer vision, driven by the increasing demand for semantic understanding of 3D environments in autonomous driving (Zheng et al., 2024; Chen et al., 2024b) and robotics (Huang et al., 2023; Chen et al., 2024c). This task aims to enable users to interact with complex 3D environments using free-form natural language queries (Guo et al., 2024), thereby narrowing the gap between human cognition and computational systems. Various methods have been proposed to advance open-vocabulary 3D scene understanding by leveraging unified 3D scene representations (Peng et al., 2023; Liu et al., 2023; Kerr et al., 2023). 3D Gaussian Splatting (3DGS) (Kerbl et al., 2023) has gained prominence as a representative

method for 3D scene construction, primarily due to its capability for real-time rendering and its inherently differentiable formulation.

As a prevailing paradigm for 3D scene understanding via 3DGS, LangSplat (Qin et al., 2024) and its successors (Qu et al., 2024; Peng et al., 2025; Li et al., 2025) have made significant strides by embedding 2D visual features from models like CLIP (Radford et al., 2021) into 3D Gaussians. This paradigm supports semantic querying at the object category level (*e.g.*, "mug," "apple"). However, this approach is fundamentally limited to a **coarse-grained** understanding, a limitation reinforced by benchmarks that primarily feature simple category-level queries (Kerr et al., 2023). We argue that real-world interaction demands a more sophisticated capability, which we define as the task of **Fine-grained Open-vocabulary 3D Scene Understanding**. Specifically, this task requires intelligent systems to move beyond category-level understanding to perceive and reason about a rich spectrum of fine-grained attributes, including visual properties (*e.g.*, color, shape, texture, material), functionality, spatial relationships, and so on. These capabilities are essential for enabling fine-grained understanding and precise querying of 3D scenes.

The key challenge of fine-grained 3D scene understanding lies in the limitations of modeling vision-language (VL) feature fields that capture rich, fine-grained semantics. One *de facto* paradigm is to leverage VL models like CLIP (Radford et al., 2021) to extract semantically-aligned 2D visual features and lift them into 3D Gaussian representations to construct a language feature field (Qin et al., 2024; Li et al., 2025); refer to §2 for more discussion. However, models like CLIP often exhibit a "bag-of-words (BoW)" behavior (Yüksekgönül et al., 2023; Koishigarina et al., 2025), interpreting images and text as sets of individual concepts while overlooking the attribute bindings between objects. For instance in Figure 1, existing methods exhibit feature mismatching among multiple attributed objects when handling descriptive queries like "`a glass with blue cap on the table`", causing the erroneous 3D object localizations. As a result, the model's representational capacity makes it difficult to capture multi-dimensional fine-grained semantics in 3D scenes.

To address the above challenge, we propose **FineSplat**, a novel 3D language Gaussian splatting framework tailored for fine-grained open-vocabulary 3D scene understanding. Instead of teaching 3D Gaussians to replicate CLIP's visual features, FineSplat constructs a language feature field solely from the language modality, thereby reformulating the problem from a noisy vision-to-text matching task into a robust *intra-modal text retrieval task*: matching a concise user query against rich, detailed captions (§3.4). As a result, the process is grounded in the intrinsic consistency of language representations, which effectively alleviates the issue of fine-grained semantic mismatch. Specifically, we devise two key strategies in FineSplat: ❶ **Fine-Grained Caption Generation (FGCG)**, which leverages an advanced caption model and Multimodal Large Language Models (MLLMs) to produce detailed, multi-attribute descriptions for every object in multi-view images; and ❷ **Fine-Grained Feature Field Modeling (FGFFM)**, which encodes these rich captions into 3D semantic features that subsequently supervise the training of the Gaussian representations.

As for evaluation, prior benchmarks such as LERF (Kerr et al., 2023) and 3D-OVS (Liu et al., 2023) are ill-suited for this task in that the queries are simple category labels, leading to performance saturation that masks the nuanced capabilities of advanced models. To address this critical gap, we construct **Fine-OVS**, a benchmark comprising 8 scenes with queries annotated across 8 fine-grained attribute types, including category label, 4 visual attributes (color, shape, texture, material), functionality, spatial relationship, and modifier, thus enabling a comprehensive evaluation of model performance in fine-grained scene understanding. Extensive experiments on Fine-OVS establish the following advantages of our method.

▷ **Fine-grained 3D Understanding.** FineSplat achieves 90.1% accuracy and 57.0% mIoU in object localization and semantic segmentation, surpassing prior arts by 21.6% and 10.3%, respectively, demonstrating a superior ability to ground complex queries involving fine-grained parts, functionality, and spatial relationships, where prior CLIP-based models consistently perform unsatisfactorily.

▷ **Discriminative Features Rendering.** The proposed FineSplat is capable of modeling fine-grained feature fields and rendering discriminative fine-grained features, whereas competing methods are limited to coarse-grained feature rendering.

▷ **Scarce View Handling.** To simulate more practical, sparse-view setting, we introduce Fine-OVS-Sparse by drastically reducing number of views down to 10%, in which FineSplat maintains its superior performance over all competing methods.

These results confirm the effectiveness of our proposed framework for the challenging task of fine-grained open-vocabulary 3D scene understanding.

## 2 RELATED WORK

**3D Gaussian Splattion**   Recently, 3D Gaussian Splatting (3DGS) (Kerbl et al., 2023) has attracted broad attention for its efficient training and faster rendering compared to NeRF (Mildenhall et al., 2020). Beyond 3D reconstruction (Yu et al., 2024; Wang et al., 2025; Tang et al., 2025), numerous studies have applied it to various downstream tasks. DreamGaussian (Tang et al., 2023) adapts 3DGS for generative tasks, achieving fast and high-quality 3D content creation from text or images through progressive Gaussian densification and mesh-based refinement. Leveraging its explicit representation, GaussianEditor (Chen et al., 2023) and subsequent works (Qu et al., 2025) integrate 2D diffusion models for semantically guided and spatially controllable 3D editing. In addition, recent studies (Qin et al., 2024; Peng et al., 2025; Zhou et al., 2024) incorporate semantic features into 3D Gaussians to support open-vocabulary scene understanding, enabling downstream tasks such as 3D object localization and 3D semantic segmentation. This paper focuses on fine-grained open-vocabulary scene understanding tasks.

**3D Language Feature Field**   Constructing a 3D language feature field that supports natural language queries provides a viable way for open-vocabulary 3D scene understanding tasks. LERF (Kerr et al., 2023) distills CLIP features into a NeRF-based 3D scene representation. Recent efforts (Dai et al., 2025; Ye et al., 2024; Shi et al., 2024) have shifted toward scene representations based on 3D Gaussian Splatting, which enables faster rendering, with representative works including LangSplat (Qin et al., 2024) and Feature3DGS (Zhou et al., 2024). Semantic Gaussians (Guo et al., 2024) distill 2D pixel-level embeddings into 3D Gaussian primitives and further leverage a 3D sparse convolutional network. GOI (Qu et al., 2024) employs feature clustering and an optimizable semantic hyperplane to enable efficient and accurate 3D open-vocabulary querying. GAGS (Peng et al., 2025) leverages SAM's prompt point density and an unsupervised granularity factor to distill multiview-consistent 2D features into 3D scenes. LangSplatV2 (Li et al., 2025) extends LangSplat by introducing high-dimensional semantic fields, leading to improved performance and faster inference. The above methods follow a common pipeline: during training, 2D visual features are extracted using a vision encoder and lifted into 3D Gaussian representations to construct a feature field; during evaluation, a text encoder extracts features from a given query, which are then matched with the visual features in the feature field based on similarity. However, existing methods mainly focus on coarse-grained scene understanding, where the construction of language feature fields is limited to object categories. In contrast, this paper addresses the more challenging task of fine-grained scene understanding by constructing feature fields enriched with fine-grained semantic attributes.

**Foundation Models**   In recent years, foundation models have exhibited remarkable capability across vision, language, and multimodal domains. SAM (Kirillov et al., 2023; Ravi et al., 2025) is capable of producing high-quality 2D segmentation masks in the zero-shot setting. The vision-language modules, such as CLIP (Radford et al., 2021), SigLIP (Zhai et al., 2023), and LSeg (Li et al., 2022), align the visual and textual modalities by embedding images and text into a unified feature space. DAM (Lian et al., 2025) uses focal prompts and a localized vision backbone to generate detailed captions for specific image and video regions. Multimodal Large Language Models (MLLMs) like GPT-4o (Hurst et al., 2024) and Qwen-VL (Bai et al., 2025) further integrate visual and textual features to perform complex reasoning and generate context-aware multimodal outputs.

## 3 FINE-GRAINED 3D LANGUAGE GAUSSIAN SPLATTING

The overall framework of our method is presented in Figure 2. Given multi-view RGB images $\mathbf{I} = \{\mathbf{I}_t \mid t = 1, 2, \ldots, T\}$ and their corresponding camera poses, we first train 3DGS to reconstruct the RGB 3D scene. We then leverage the proposed Fine-Grained Caption Generation (FGCG) strategy to obtain textual descriptions that include the objects' multi-dimensional fine-grained attributes. Finally, we introduce the Fine-Grained Feature Field Modeling (FGFFM) module to construct a feature field enriched with fine-grained semantics. During the inference stage, given any fine-grained open-vocabulary query, we encode it using a text encoder and compute its similarity with the features decoded from the 2D rendered feature to produce the localization and segmentation results. We provide detailed descriptions of each component of our framework in this section.

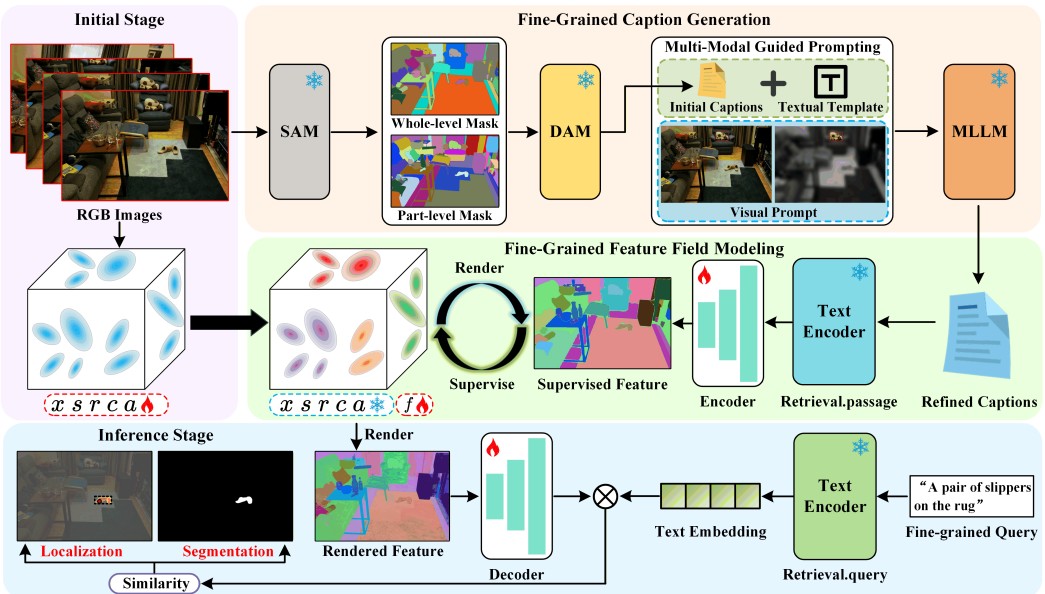

Figure 2: **The overall framework of our FineSplat.** The model consists of four main components: the initial stage, the Fine-Grained Caption Generation strategy, the Fine-Grained Feature Field Modeling strategy, and the inference stage.

## 3.1 PRELIMINARIES

3D Gaussian Splatting (3DGS) models a 3D scene using a set of anisotropic Gaussian ellipsoids that explicitly encode spatial position, orientation, and scale. Each Gaussian $G(x)$ is formulated as

$$G(x) = \exp(-\frac{1}{2}(x - \mu)^\top \Sigma^{-1}(x - \mu)), \tag{1}$$

where $x \in \mathbb{R}^3$ is the centroid, $\mu \in \mathbb{R}^3$ denotes the mean, and $\Sigma$ is a positive semi-definite covariance matrix computed from the rotation matrix and the scaling matrix. During rendering, Gaussian primitives are first projected onto the 2D image plane. We adopt a splatting-based formulation to compute the covariance matrix $\Sigma'$ in camera coordinates:

$$\Sigma' = JW\Sigma W^T J^T, \tag{2}$$

where $J$ denotes the Jacobian matrix of the projective transformation and $W$ is the world-to-camera transformation matrix. After obtaining the projected Gaussian primitives, 3DGS renders the final image using a point-based $\alpha$-blending process to compute the color $C$ of each pixel $v$:

$$C(v) = \sum_{i \in \mathcal{N}} c_i \alpha_i \prod_{j=1}^{i-1}(1 - \alpha_j), \tag{3}$$

where $\mathcal{N}$ denotes the set of 3D Gaussians sorted by front-to-back depth, $\alpha_i$ is the opacity of the $i$-th point, and $c_i$ is its color computed using spherical harmonics.

## 3.2 FINE-GRAINED CAPTION GENERATION

As discussed above, current 3D scene understanding methods (Qin et al., 2024; Peng et al., 2025; Li et al., 2025) typically leverage visual features from pretrained vision–language aligned models such as CLIP to construct 3D Gaussian representations. However, these methods often suffer from vision-text feature mismatching, resulting in erroneous outcomes when handling fine-grained queries. In contrast, single-modal features inherently exhibit strong internal consistency. Motivated by these analyses, we design Fine-Grained Caption Generation (FGCG) strategy that transforms object visual features into descriptive captions enriched with multi-dimensional fine-grained attributes, thereby enhancing feature matching for fine-grained semantics.

There are two main approaches to obtaining image captions: one leverages foundation models specifically designed for caption generation, while the other exploits Multimodal Large Language Models (MLLMs) with strong understanding and reasoning capabilities. However, both approaches have limitations: Well-trained caption models (Lian et al., 2025) can provide detailed and precise descriptions of object appearance but lack spatial relationships and functionality that are crucial for fine-grained 3D scene understanding; MLLMs (Hurst et al., 2024; Bai et al., 2025) often lead to erroneous or incomplete attribute descriptions due to their broad training targets and intrinsic hallucination problem (Liu et al., 2024; Sahoo et al., 2024). Therefore, the proposed FGCG strategy operates in two stages: first, a caption generation model produces an initial caption, and subsequently, an MLLM refines and completes the caption that provides a relatively precise start point.

**Caption Generation with Semantic Mask**   Specifically, given an input image $\mathbf{I}_t \in \mathbb{R}^{3 \times H \times W}$, where $H$ and $W$ denote the image height and width, we first employ SAM (Kirillov et al., 2023) to obtain high-quality whole-object masks $\mathbf{M}^w$ and part-level masks $\mathbf{M}^p$, with the goal of generating detailed descriptions for each object and its constituent parts.

To describe the visual content within each mask, we adopt DAM (Lian et al., 2025), which is capable of producing accurate and detailed captions for specified image regions. For each mask, DAM generates an initial caption $C$ for the corresponding object, capturing both its category and visual attributes, as follows,

$$C_{l,k} = \text{DAM}(\mathbf{M}_k^l), \quad l \in \{w, p\}, \tag{4}$$

where $l$ denotes the mask level, and $k$ denotes the index of the object within that level.

**Multi-Modal Guided Prompting (MMGP)**   As analyzed above, the initial captions remain insufficient for fine-grained 3D scene understanding, so we propose a Multi-Modal Guided Prompting (MMGP) submodule to further refine the initial captions.

Specifically, MMGP integrates both textual and visual inputs as joint inputs to the MLLM. The textual prompt is formed by combining the initial captions with a textual template T (refer to Appendix E.1 for template detail), guiding the MLLM to refine the original captions more precisely and supplement missing fine-grained attribute descriptions. The visual input adopts a dual-image design: one image $\mathbf{I}^{\text{highlight}}$ highlights the target object with a red bounding box to emphasize its spatial relationships within the environment, while the other $\mathbf{I}^{\text{blur}}$ applies Gaussian blur to all non-target regions, directing the MLLM's focus to the object itself. The refined fine-grained caption $\hat{C}$ can be obtained through the above process:

$$\hat{C}_{l,k} = \text{MLLM}\left(\text{Textual}(C_{l,k} \cup \text{T}), \text{Visual}(\mathbf{I}_t^{highlight} \cup \mathbf{I}_t^{blur})\right). \tag{5}$$

The resulting $\hat{C}$ contains multi-dimensional fine-grained attributes, providing rich semantic guidance for constructing the fine-grained feature field. Further details are provided in the Appendix E.3.

### 3.3 Fine-Grained Feature Field Modeling

After obtaining object-wise fine-grained captions in the previous stage, extracting object-level features is essential for constructing a high-quality feature field that encodes rich, fine-grained semantics, thereby facilitating fine-grained 3D scene representation construction and understanding.

The captions $\hat{C}$, providing a comprehensive description of the objects' multi-dimensional fine-grained attributes, correspond to a *long passage*, whereas the inference queries, containing only a subset of attributes, correspond to a *short query*. We treat this process as a *asymmetric text retrieval* task, in which the intrinsic coherence of language representations effectively mitigates fine-grained semantic mismatches.

In asymmetric retrieval tasks, it is common to employ two encoders: a passage encoder to encode long passages and a query encoder to encode short queries. Accordingly, we choose the passage encoder to extract the language features for long captions. The feature field training of 3DGS requires supervision with pixel-aligned features; therefore, we assign the caption features to all pixels within the same mask region corresponding to an object or a part. The whole ($w$) and part ($p$) semantic level pixel-aligned language features $\mathbf{F}_t^w, \mathbf{F}_t^p \in \mathbb{R}^{H \times W \times D}$ for the input image $\mathbf{I}_t$ as follows,

$$\mathbf{F}_t^l(v_k) = E_{passage}(\hat{C}_{l,k}) \in \mathbb{R}^D, l \in \{w, p\}, \tag{6}$$

where $v_k$ represents any pixel within $\mathbf{M}_k^l$, $E_{passage}$ denotes the text encoder in passage mode.

**Lightweight Autoencoder** Since language features $\mathbf{F}_t$ are high-dimensional, directly training them in scenes with millions of Gaussians would significantly increase training time and memory overhead. We train a lightweight autoencoder for each scene to further compress the high-dimensional features. Specifically, the encoder $\phi$ first maps the high-dimensional features into a low-dimensional space $\mathbb{R}^d$, where $d \ll D$, and then the decoder $\psi$ is designed to reconstruct the original high-dimensional language embeddings. The optimization objective is defined as follows,

$$\mathcal{L}_{\text{auto}} = \sum_{l \in \{w,p\}} \sum_{t=1}^{T} \mathcal{L}\left(\psi\left(\phi\left(\mathbf{F}_t^l(v)\right)\right), \mathbf{F}_t^l(v)\right). \tag{7}$$

Then, we adopt a volume rendering process similar to color rendering to train the low-dimensional features $\{\boldsymbol{f}^w, \boldsymbol{f}^p\}$ of each Gaussian primitive, which can be formulated as

$$\hat{\mathbf{F}}^l(v) = \sum_{i \in \mathcal{N}} \boldsymbol{f}_i^l \alpha_i \prod_{j=1}^{i-1}(1 - \alpha_j), l \in \{w,p\}, \tag{8}$$

where $\hat{\mathbf{F}}^l(v)$ represents the fine-grained semantic embedding rendered at pixel $(v)$, and the optimization objective of the fine-grained feature field is as follows,

$$\mathcal{L}_f = \sum_{l \in \{w,p\}} \sum_{t=1}^{T} \mathcal{L}\left(\hat{\mathbf{F}}_t^l(v), \phi(\mathbf{F}_t^l(v))\right). \tag{9}$$

Through the above process, we successfully construct a fine-grained feature field using fine-grained semantic embeddings. The resulting representation offers an effective solution for fine-grained scene understanding, enabling the model to accurately localize and segment target objects under complex fine-grained open-vocabulary queries. Further details are provided in the Appendix E.3.

## 3.4 FINE-GRAINED OPEN-VOCABULARY QUERYING

During the inference stage, given a fine-grained textual query, our goal is to retrieve the region of interest under arbitrary rendering views from the feature field constructed in Section 3.3. Therefore, we still use the same text encoder, but switch from passage mode to query mode, to obtain the embedding $\boldsymbol{f}_q \in \mathbb{R}^D$ of the fine-grained query. For each feature map rendered from the fine-grained feature field, we employ the trained decoder $\psi$ to reconstruct it into the original high-dimensional feature map $\psi(\hat{\mathbf{F}}^l) \in \mathbb{R}^{H \times W \times D}$. Next, we compute the similarity between the text embedding and each pixel in the feature map, thereby obtaining two relevance maps $\{\mathbf{R}_w, \mathbf{R}_p\}$, each corresponding to a semantic level (whole and part) defined by SAM, as follows,

$$\mathbf{R}_l = \frac{\psi(\hat{\mathbf{F}}^l) \cdot \boldsymbol{f}_q}{\|\psi(\hat{\mathbf{F}}^l)\| \|\boldsymbol{f}_q\|} \in \mathbb{R}^{H \times W}, l \in \{w,p\}. \tag{10}$$

Following the strategy used in LangSplat (Qin et al., 2024), we select the semantic level with the highest relevance score. For the 3D object localization task, the pixel with the highest relevance score is directly chosen as the result. For the 3D semantic segmentation task, pixels with relevance scores above a predefined threshold are retained as the predicted mask of the target object.

## 4 EXPERIMENT

### 4.1 FINE-OVS BENCHMARK DATASET

Existing datasets, such as LERF (Kerr et al., 2023) and Mip-NeRF360 (Barron et al., 2022), primarily employ coarse-grained object labels as evaluation queries, limiting their suitability for fine-grained scene understanding research. To address this gap, we construct **Fine-OVS**, a novel benchmark dataset specifically designed for the fine-grained scene understanding task. Fine-OVS contains 8 scenes and 43 test frames in total. Among them, *teatime*, *ramen*, *kitchen*, and *figurines* are sourced from LERF, while *room*, *counter*, *bonsai*, and *garden* are sourced from Mip-NeRF360. We define 8 fine-grained attributes (category label, color, shape, texture, material, functionality, spatial relationship, and modifier) and annotate 500 test queries accordingly. To characterize the composition of the benchmark queries, we further analyze the frequency distribution of these attributes, as shown in Figure 3. Further details of Fine-OVS, along with some cases, are provided in Appendix C.

Table 1: Quantitative comparisons of **3D object localization** on the Fine-OVS. We report the mean accuracy(%). The highest results are highlighted, while the second-highest results are underlined.

| Method | Fine-OVS | | | | | | | | |
|---|---|---|---|---|---|---|---|---|---|
| | Teatime | Ramen | Kitchen | Figurines | Room | Counter | Bonsai | Garden | Overall |
| GS-Grouping | 42.3 | 48.2 | 47.1 | 42.6 | 54.5 | 52.1 | 51.9 | 39.5 | 47.3 |
| GAGS | 66.7 | 67.5 | 60.8 | 67.6 | 66.6 | 74.6 | 73.0 | 71.0 | 68.5 |
| LangSplat | 63.8 | 73.5 | 41.1 | 73.4 | 68.2 | 64.8 | 69.2 | 50.0 | 63.0 |
| LangSplatV2 | 69.5 | 77.1 | 49.0 | 70.6 | 63.6 | 64.8 | 69.2 | 57.9 | 65.2 |
| Ours | **87.0** | **89.2** | **90.2** | **89.7** | **89.4** | **92.9** | **90.3** | **92.1** | **90.1** |

Table 2: Quantitative comparisons of **3D semantic segmentation** on the Fine-OVS. We report the mean IoU(%). The highest results are highlighted, while the second-highest results are underlined.

| Method | Fine-OVS | | | | | | | | |
|---|---|---|---|---|---|---|---|---|---|
| | Teatime | Ramen | Kitchen | Figurines | Room | Counter | Bonsai | Garden | Overall |
| GS-Grouping | 40.9 | 41.3 | 31.1 | 35.5 | 46.0 | 41.3 | 28.5 | 28.5 | 36.6 |
| GAGS | 42.7 | 46.5 | 32.4 | 41.9 | 45.4 | 57.1 | 41.2 | 41.2 | 43.6 |
| LangSplat | 39.9 | 46.4 | 21.6 | 43.1 | 42.7 | 50.6 | 34.3 | 34.3 | 39.1 |
| LangSplatV2 | 47.0 | 53.8 | 33.5 | 50.9 | 48.2 | 52.9 | 43.7 | 43.7 | 46.7 |
| Ours | **54.2** | **61.6** | **53.1** | **53.4** | **62.4** | **64.3** | **53.3** | **53.3** | **57.0** |

## 4.2 IMPLEMENTATION DETAILS

We employ the SAM ViT-H (Kirillov et al., 2023) model to generate high-quality 2D masks. The fine-grained caption corresponding to each mask object is generated by the DAM-3B (Lian et al., 2025) and Qwen2.5-VL-7B-Instruct (Team, 2025) models. The jina-embeddings-v3 model (Sturua et al., 2024) is employed as the text encoder to encode fine-grained captions and fine-grained queries, with an output dimension of 1024. When modeling the fine-grained feature field, the dimension of the Gaussian language feature is set to 6. All our experiments are conducted on a single 24G RTX-4090 GPU. More implementation details are provided in the Appendix B.

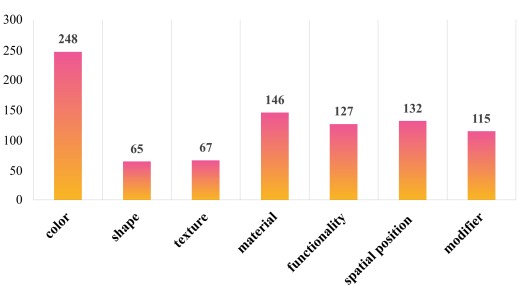

Figure 3: Frequency distribution of 7 fine-grained attributes in the Fine-OVS (excluding the category labels present in nearly all queries).

## 4.3 QUANTITATIVE COMPARISONS WITH BASELINE METHODS

We compared the proposed FineSplat with the current state-of-the-art scene understanding methods, including GS-Grouping (Ye et al., 2024), GAGS (Peng et al., 2025), LangSplat (Qin et al., 2024), and LangSplatV2 (Li et al., 2025), on the Fine-OVS dataset. The experimental results for 3D object localization and 3D semantic segmentation are shown in Tables 1 and 2, respectively.

Based on the experimental results, it can be observed: Our method achieves significant improvements in both localization accuracy and segmentation IoU score across all scenes. Overall, the localization accuracy increases by 21.6% and the segmentation IoU score by 10.3%, demonstrating the effectiveness of FineSplat. Moreover, compared to other scenes, the performance of the baseline methods drops significantly in the relatively complex *Kitchen* and *Garden* scenes. In contrast, FineSplat maintains high localization accuracy and segmentation IoU, demonstrating its stability and strong fine-grained understanding capability in challenging scenes.

Table 3: Quantitative comparisons of **3D semantic segmentation** on the Fine-OVS-Sparse.

| Method | Fine-OVS-Sparse | | | | |
| | Teatime | Ramen | Kitchen | Figurines | Overall |
|---|---|---|---|---|---|
| GAGS | 38.5 | 28.4 | 8.8 | 23.4 | 24.8 |
| LangSplatV2 | 39.5 | 40.2 | 17.3 | **29.2** | 31.6 |
| Ours | **45.3** | **44.8** | **36.7** | 27.6 | **38.6** |

Table 4: Ablation Study on Components of the FGCG Strategy.

| Component | | Teatime | | Ramen | | Kitchen | | Figurines | |
| DAM | MLLM | Acc (%) | IoU (%) | Acc (%) | IoU (%) | Acc (%) | IoU (%) | Acc (%) | IoU (%) |
|---|---|---|---|---|---|---|---|---|---|
| ✓ | | 78.2 | 48.1 | 80.5 | 55.8 | 62.7 | 44.9 | 81.2 | 48.9 |
| | ✓ | 52.2 | 30.0 | 67.5 | 40.3 | 54.9 | 30.5 | 79.4 | 39.5 |
| ✓ | ✓ | **91.3** | **54.4** | **89.2** | **61.6** | **90.2** | **53.1** | **89.7** | **53.4** |

**Sparse-view Setting**    In real-world applications such as autonomous driving, AR/VR, and robotic navigation, 3D scene understanding often relies on only a limited set of viewpoints (Chen et al., 2024a). To better reflect these practical constraints, we introduce a sparse-view setting, Fine-OVS-Sparse, in which the number of training views is reduced to just 10%. We evaluate both state-of-the-art methods and our proposed method under this setting, with the quantitative results reported in Table 3. The results show that, even under sparse-view conditions, our method consistently outperforms competing methods. For additional analyses and comparisons, please refer to the Appendix D.

## 4.4 ANALYSES

**Ablation Study of the FGCG**    To further demonstrate that FGCG is an ideal strategy for generating object captions with rich fine-grained attributes, we conduct an ablation study on its two core components, DAM and MLLM, with the results shown in Table 4. The results indicate that using DAM alone achieves reasonably strong performance but falls short of the full FGCG strategy because it does not model spatial relationships or functionality, both critical for fine-grained scene understanding. MLLM alone captures some detailed fine-grained attributes, but without initial captions, its performance is limited, most evidently reflected in the IoU scores. By first generating initial captions with DAM and subsequently refining them using MLLM, FGCG generates detailed multi-dimensional fine-grained descriptions and achieves the best overall performance.

We also conduct an additional ablation study of the Gaussian language feature dimension. Due to the space limitation, refer to the Appendix E.4 for details. We also provide experiments and analyses on the generalization capability of FineSplat in Appendix E.7.

**Qualitative Comparisons of Feature Maps**    To further illustrate that the feature field modeled by FineSplat encodes rich fine-grained semantic information, we present qualitative comparisons of feature maps rendered from the feature fields produced by different methods, as shown in Figure 4. The results reveal that existing methods remain limited to coarse-grained feature field modeling: they can distinguish only objects with substantial visual differences and lack the ability to differentiate fine-grained targets, such as the various ingredients within a bowl of *ramen*. In contrast, the feature field constructed by our method effectively captures these subtle distinctions, enabling a more precise and fine-grained understanding of complex scenes.

**Qualitative Comparisons of 3D Object Localization**    Figure 5 shows the visualization results of fine-grained open-vocabulary 3D object localization. It can be observed that LangSplat, GAGS, and LangSplatV2 fail to effectively activate the target object regions when handling fine-grained queries, resulting in erroneous localization due to feature mismatches. In contrast, our method can accurately respond to fine-grained queries, not only at the part level (*e.g.*, "a round bear nose") but also for more challenging abstract descriptions involving spatial relationships (*e.g.*, "an object used for vision correction on the table"). We conduct further qualitative visualization

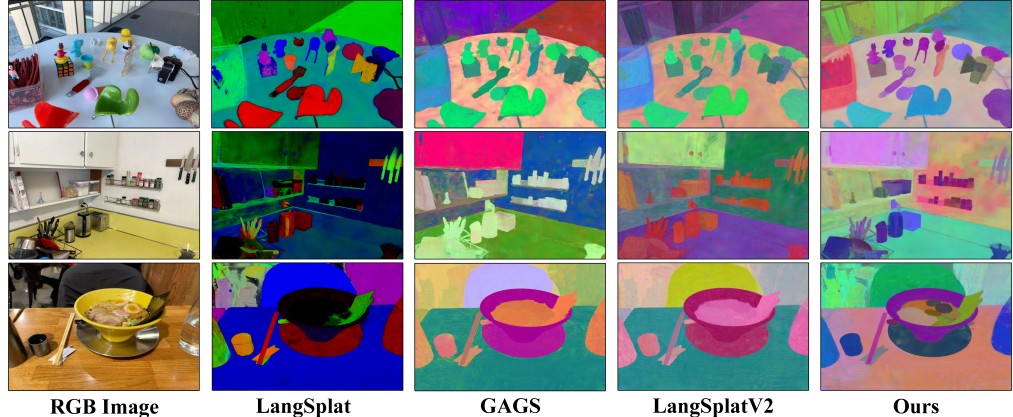

Figure 4: Qualitative comparisons of feature maps rendered from the trained feature field, where closer colors reflect more similar features.

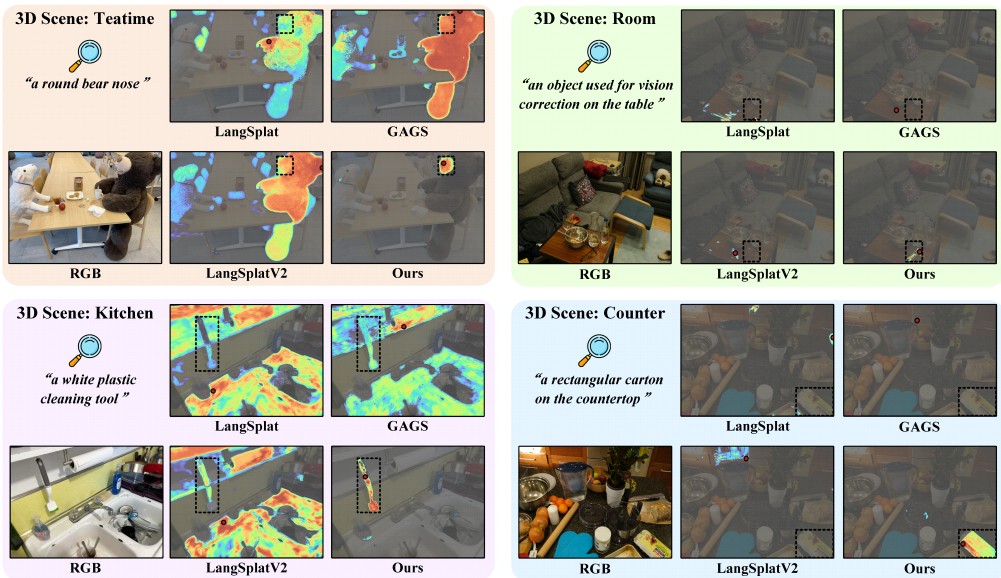

Figure 5: Qualitative comparisons of **fine-grained open-vocabulary 3D object localization** on the Fine-OVS dataset. Black dashed boxes denote the annotated regions, and red points correspond to the predictions of the model.

comparisons of fine-grained open-vocabulary 3D object localization and 3D semantic segmentation. Please refer to the Appendices E.8 and E.9 for details.

## 5 CONCLUSION

In this paper, we introduce a more challenging task of **Fine-grained Open-vocabulary 3D Scene Understanding** and propose a novel fine-grained 3D language gaussian splatting, **FineSplat** for short. FineSplat integrates two main strategies: the FGCG strategy, which produces object captions enriched with multi-dimensional fine-grained attributes, and the FGFFM strategy, which constructs a fine-grained language feature field for precise scene representation. To support research and evaluation on this task, we further construct **Fine-OVS**, a dedicated benchmark comprising diverse scenes and fine-grained queries. Extensive experiments on Fine-OVS demonstrate that FineSplat consistently and significantly outperforms existing state-of-the-art methods, highlighting its effectiveness and robustness for fine-grained understanding of complex 3D scenes.

ETHICS STATEMENT

We have thoroughly reviewed the ICLR Code of Ethics and confirm that all aspects of our work comply with established academic ethical standards. Our research does not involve human or animal subjects, nor does it contain any potentially harmful insights, methodologies, or applications. We do not anticipate any issues related to discrimination, bias, fairness, privacy, or security. Our benchmark data are adapted from existing datasets, so there are no license issues. In addition, we recruited full-time adults for data annotation and provided payment. Furthermore, our work adheres to all relevant legal and research integrity requirements, and we are confident that it aligns with the principles outlined in the ICLR Code of Ethics.

REPRODUCIBILITY STATEMENT

We have made every effort to ensure that our work is reproducible. All essential technical details, model configurations, and quantitative results are provided in the main paper and the Appendix. We will release our annotated dataset and source code, along with detailed instructions. We believe this will help the community verify our results and build upon them.

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

# Appendices

**Use of Large Language Models (LLMs).** We used a large language model solely as a writing aid to improve the clarity, grammar, and overall readability of the manuscript. Its role was limited to polishing the language and refining sentence structure, without contributing to research ideation, experimental design or data analysis. All technical ideas, methods, results, and conclusions are entirely the work of the authors, and we take full responsibility for the final content.

## A  MORE RELATION WORK

In the vision domain, DINO (Caron et al., 2021) and DINOv2 (Oquab et al., 2023) learn high-quality visual representations from unlabeled images through self-supervised learning. For the language-specific tasks, Large Language Models (LLMs) such as LLaMA (Touvron et al., 2023; Dubey et al., 2024) and Qwen (Yang et al., 2025a; 2024) series have demonstrated remarkable improvements in inferential capabilities. GPT-oss (OpenAI et al., 2025) leverages Mixture-of-Experts architectures to further improve training and inference efficiency.

Given the remarkable progress of 2D foundation models in visual and multimodal understanding, a growing line of work (Koch et al., 2025; Chen et al., 2022; Yang et al., 2025b) seeks to repurpose their learned representations for 3D tasks, thereby opening new frontiers in 3D perception, reconstruction, and reasoning.

## B  MORE IMPLEMENTATION DETAILS

Our method is implemented based on the official codebase of 3D Gaussian Splatting (3DGS) (Kerbl et al., 2023) and LangSplat (Qin et al., 2024). Follow the default parameter setting as 3DGS, we train each scene for 30,000 iterations in the initial stage. For the Fine-Grained Feature Field Modeling strategy, We train only the language features for 30,000 iterations. The scene special autoencoder is implemented by an MLP. During the fine-grained open-vocabulary querying stage, for 3D semantic segmentation, we first obtain the maximum value in the activation map and set the filtering threshold to 0.88 of this maximum. This threshold is applied uniformly in experiments across all scenes.

For dataset preparation of Fine-OVS-Sparse, we construct a lightweight variant of the Fine-OVS dataset. Specifically, for each of the four LERF scenes, we manually downsample the available training images by a factor of 10. During this process, evaluation viewpoints as well as low-quality or blurry images were deliberately excluded to ensure a clean training set while preserving representative scene coverage.

Table 5: Statistics of the LERF, Mip_Nerf360, and the proposed Fine-OVS.

| Dataset | Scene | Test Frame | Attribute | Test Query | | | | |
|---|---|---|---|---|---|---|---|---|
| | | | | 1 attribute | 2 attributes | 3 attributes | 4 attributes | Total |
| LERF | 4 | 22 | 3 | 157 | 52 | 0 | 0 | 209 |
| Mip_Nerf360 | 4 | 17 | 4 | 40 | 89 | 4 | 0 | 133 |
| Fine-OVS | 8 | 43 | 8 | 0 | 170 | 268 | 62 | 500 |

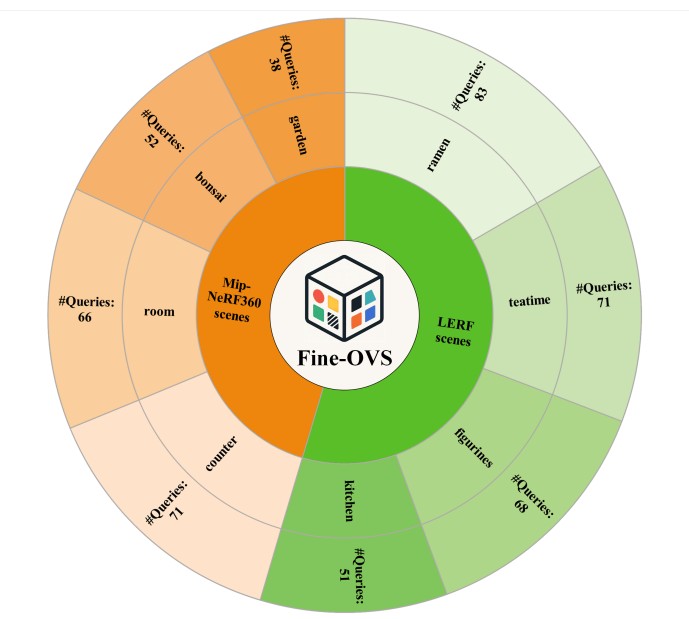

Figure 6: Distribution of queries across different scenes in the Fine-OVS dataset. The inner ring indicates the source of scenes (LERF and Mip-NeRF360), while the outer ring shows individual scenes and the number of queries annotated by us for each scene.

## C   MORE DETAILS OF FINE-OVS BENCHMARK

### C.1   COMPREHENSIVE STATISTICS OF FINE-OVS

We present more statistics of the Fine-OVS benchmark dataset. Specifically, as shown in Table 5, the proposed Fine-OVS dataset consists of 8 scenes with a total of 43 test frames. The test queries consist of 2 to 4 fine-grained attributes, with 170 queries containing 2 attributes, 268 queries containing 3 attributes, and 62 queries containing 4 attributes. Compared with the LERF and Mip_Nerf360 datasets designed for coarse-grained scene understanding, Fine-OVS exhibits significant improvements in multiple aspects, including the number of scenes, the diversity of attributes, the scale of queries, and, most importantly, the complexity of the queries. Additionally, Figure 6 illustrates the distribution of queries across the different scenes.

### C.2   SEMI-AUTOMATIC ANNOTATION OF FINE-OVS

For any selected test frame, we first use SAM (Kirillov et al., 2023) to obtain the bounding box and segmentation mask for each target object. The bounding box, together with the test frame, is then fed into DAM (Lian et al., 2025) to generate the object caption, which includes the semantic category, four visual attributes, and modifier. Next, we manually annotate each object's functionality and spatial position, resulting in a complete eight-dimensional fine-grained attributes. Finally, 2 to 4 attributes are manually selected from each object and combined into a phrase, which is used as an annotation query for evaluating the fine-grained open-vocabulary scene understanding task.

| Test Frame | Test Frame | Test Frame |
|---|---|---|
| **Test Queries** | **Test Queries** | **Test Queries** |
| 1. a spoon soaked in liquid | 1. a green plastic object | 1. a light gray ceramic object used for holding plants |
| 2. a dark brown toy used for comfort | 2. white refrigerator door | 2. a blue dishtowel |
| 3. a white toy used for comfort | 3. a metallic pot hanging on the wall | 3. a yellow dishtowel |
| 4. a red smooth food on the table | 4. a red spatula used for cooking | 4. a smooth metal bowl |
| 5. a brown bag used for holding items | 5. a beige spoon | 5. a transparent plastic lid on the countertop |
| 6. a white ceramic mug | 6. a smooth wooden table | 6. a plant with dark green leaves |
| 7. a clear glass on the table | 7. a round metal object with fine mesh | 7. a red can on the countertop |
| 8. a brown bear nose and a white bear head | 8. a pair of scissors used for cutting hanging on the wall | 8. a wooden object used for flattening dough |
| 9. a white plate on the table | 9. a metal grater used for grating food | 9. oranges used for eating |
| 10. brown and crispy food on the plate | 10. a cylindrical ceramic flowerpot | 10. an oven control panel showing "12:21" |
| 11. a blue ribbon with dall-e brand | | 11. a blue silicone object |
| 12. a soft paper used for cleaning | | 12. a pitcher used for holding liquid |

Figure 7: Examples of test frames and their corresponding test queries in the Fine-OVS dataset.

## C.3 TEST QUERIES IN FINE-OVS

To more intuitively present the test queries of Fine-OVS, we show in Figure 7 the test queries corresponding to several test frames. It can be observed that the queries in the test frames go beyond simple category labels, incorporating multi-dimensional fine-grained information such as visual attributes, functionality, and spatial relations. These more challenging queries enable Fine-OVS to comprehensively evaluate different methods on the fine-grained scene understanding task, thereby better simulating the demands of real-world applications.

## D MORE QUANTITATIVE COMPARISONS

### D.1 SPARSE-VIEW SETTING AND PLUG-IN-BASED ENHANCEMENT

In this section, we first analyze the reasons behind the performance degradation observed in Table 3 under the sparse-view setting, and then introduce a plug-in-based general module to alleviate this challenge. Specifically, we attribute the performance drop primarily to two factors:

- **Degraded scene reconstruction:** As shown in Figure 8, sparse-view constraints yield weaker geometric and photometric priors, leading to suboptimal scene initialization and impairing subsequent semantic learning.
- **Reduced cross-view semantic consistency:** Significant appearance variations across viewpoints introduce discrepancies in 2D semantic features, undermining cross-view correspondences and hindering coherent multi-view representation learning.

To this end, we introduce a viewpoint augmentation strategy designed to alleviate the challenges introduced by sparse-view constraints. Specifically, we first employ the feed-forward multi-view reconstruction model Mast3r-SfM (Duisterhof et al., 2025) to obtain a dense point cloud initialization. We then train 3DGS for 15K iterations using the sparse viewpoints. To further enrich the training views, we interpolate between the input sparse viewpoints using multi-dimensional B-splines, thereby generating new observation camera poses. To avoid implausible viewpoints (*e.g.*,

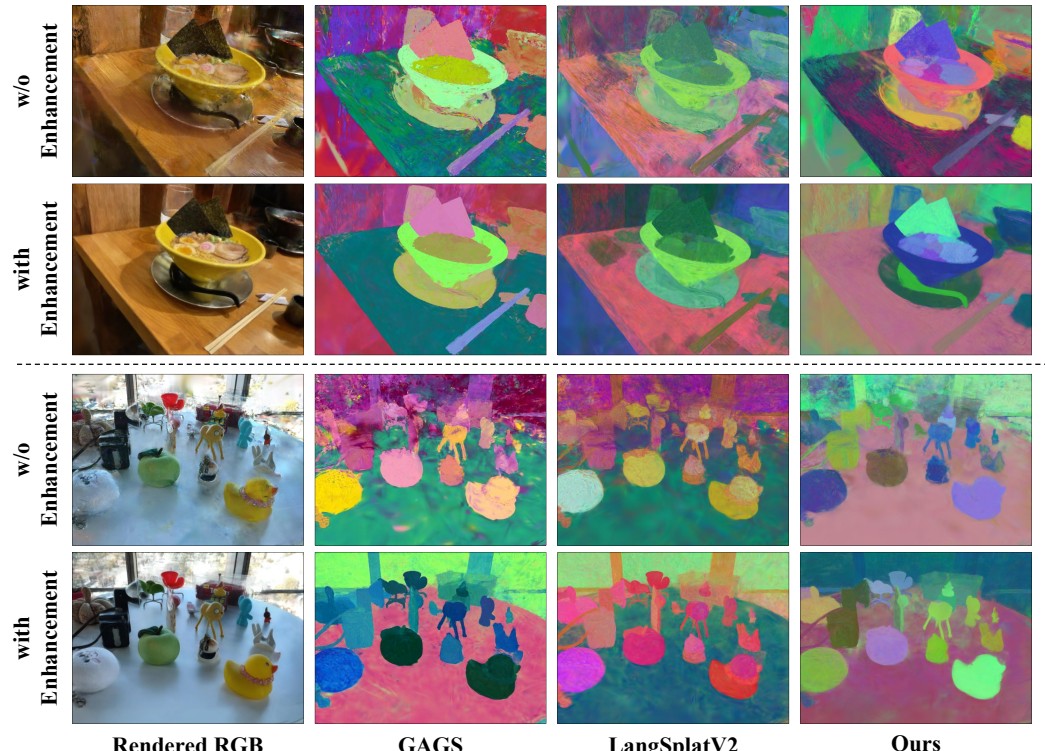

Figure 8: Qualitative comparisons of feature maps rendered from novel views before and after integrating our plug-in-based enhancement module. The top two rows correspond to the *Ramen* scene, while the bottom two rows correspond to the *Figurines* scene.

Table 6: Quantitative comparisons of 3D semantic segmentation on the Fine-OVS-Sparse after integrating our plug-in-based enhancement module. We report the mean IoU(%). Upward arrows indicate performance improvement compared to Table 3.

| Method | Fine-OVS-Sparse | | | | |
|---|---|---|---|---|---|
| | Teatime | Ramen | Kitchen | Figurines | Overall |
| GAGS | 40.1 | 30.3 | 13.9 | 34.0 | 29.6 ↑5% |
| LangSplatV2 | 41.3 | 43.6 | 20.5 | 35.9 | 35.3 ↑4% |
| Ours | **48.7** | **51.2** | **38.1** | **51.7** | **47.4** ↑9% |

those located inside walls), we compute the density of Gaussian points within a 5 cm radius around each candidate camera and discard any pose with a density exceeding 50. For each scene, the final number of generated viewpoints is set to $N_{\text{interp}} \times (M - 1)$, where $N_{\text{interp}}$ denotes the number of interpolations between two adjacent views and $M$ is the total number of input views.

We then render images from these generated viewpoints and feed them into a one-step diffusion model, Difix3D (Wu et al., 2025), for view refinement. Notably, since Difix3D requires a real captured reference frame, we assign each interpolated view the input frame with the largest co-visible overlap. The refined views are subsequently used as additional training images to fine-tune the model for 3K iterations. We then re-render from the interpolated viewpoints and perform another 3K iterations of training. In our setting, this refinement–retraining process is conducted for only two rounds.

**Analysis** To validate the effectiveness of our proposed plug-in-based enhancement module, we integrate it into both the baseline methods and our own approach, and evaluate them on the Fine-OVS-Sparse benchmark. The results, presented in Table 6, show that incorporating this module

Table 7: Reconstruction quality before and after integrating our plug-in-based enhancement module on four scenes. We report SSIM ↑, PSNR ↑, and LPIPS ↓. The results are evaluated on all original viewpoints.

| Scene | Before | | | After | | |
|---|---|---|---|---|---|---|
| | SSIM ↑ | PSNR ↑ | LPIPS ↓ | SSIM ↑ | PSNR ↑ | LPIPS ↓ |
| Teatime | 0.59 | 17.10 | 0.42 | 0.70 | 19.09 | 0.39 |
| Ramen | 0.57 | 18.30 | 0.37 | 0.74 | 20.91 | 0.31 |
| Kitchen | 0.68 | 16.40 | 0.39 | 0.69 | 17.92 | 0.36 |
| Figurines | 0.59 | 15.15 | 0.42 | 0.70 | 17.43 | 0.35 |

Table 8: Quantitative comparisons of 3D semantic segmentation on the LERF. We report the mean IoU(%). The highest results are highlighted, while the second-highest results are underlined.

| Method | LERF | | | | |
|---|---|---|---|---|---|
| | Teatime | Ramen | Kitchen | Figurines | Overall |
| GS-Grouping | _60.9_ | 32.5 | 30.4 | 32.9 | 39.2 |
| GAGS | 60.2 | 45.5 | 53.2 | _52.8_ | 52.9 |
| LangSplat | 59.7 | 45.5 | 42.3 | 50.8 | 49.6 |
| LangSplatV2 | **69.6** | _49.8_ | **56.2** | **56.4** | **58.0** |
| Ours | 56.3 | **55.2** | _54.1_ | 49.9 | _53.9_ |

leads to substantial performance gains across all methods. We attribute these gains to improved 3D scene representations and richer supervisory viewpoints enabled by our plug-in-based enhancement module. To further evaluate its impact on reconstruction quality, we present the quantitative results in Table 7. The results demonstrate that our plug-in-based enhancement module substantially improves scene reconstruction quality, thereby facilitating more effective learning of language features within 3D environments.

### D.2 Quantitative comparisons on the LERF

To validate the generalizability of the proposed FineSplat, we conduct additional quantitative experiments on the LERF dataset, which is designed for coarse-grained scene understanding, as shown in Table 8. Although FineSplat does not achieve the highest performance on the coarse-grained scene understanding task, its overall results remain competitive. This can be explained by the following reasons: on one hand, FineSplat constructs the language feature field using captions rich in fine-grained attributes, so when a query contains only the object category, the embedding has a significant distribution difference from the modeled feature field, leading to a higher likelihood of errors. On the other hand, the number of queries in the LERF dataset is relatively limited, and the test frames primarily contain repeated annotations of a few objects, which further constrains FineSplat's performance on coarse-grained scene understanding tasks.

Moreover, the model can still perform well on the coarse-grained scene understanding task, this can be mitigated with slight modification. Specifically, this only requires replacing the passage encoder used during training and the query encoder used during inference with a unified text matching encoder. The experimental results are shown in Table 9. Therefore, FineSplat not only retains the fundamental coarse-grained query capabilities but also successfully models more complex fine-grained feature fields.

Table 9: Quantitative comparisons of 3D object localization on the LERF. We report the mean accuracy(%).

| Method | LERF | | | | |
| --- | --- | --- | --- | --- | --- |
| | Teatime | Ramen | Kitchen | Figurines | Overall |
| LangSplat_V2 | 88.1 | 71.8 | 68.2 | 82.1 | 77.6 |
| Ours | 96.6 | 69.2 | 66.8 | 78.6 | **77.8** |

> ***You are given:***
> *1. The original image where the target object is highlighted by a red bounding box.*
> *2. The version of the original image where everything outside the target object is blurred.*
> *3. The initial caption that describes this target object: {prompt}*
> ***Your task:*** *Rewrite the caption as one concise, fluent sentence that must include all of the following six elements:*
> *1. The object's name.*
> *2. The object's color.*
> *3. The object's material. (e.g., metal, plastic, ceramic, wood, paper)*
> *4. The object's texture. (e.g., crumpled, glossy, rough, soft, smooth, wrinkled, fuzzy, matte)*
> *5. The object's functional attribute, confirm what the object is used for. (e.g., cleaning, holding liquid, eating)*
> *6. The object's 3D-consistent spatial relationship. (e.g., on the chair, inside the drawer, hang on the wall)*
> *If any element is missing from the initial caption, you must infer and supplement it based on the visual content of the provided images.*
> ***Guidelines:***
> *1. Make the caption concise, natural.*
> *2. Avoid vague expressions like "next to" or "near", and avoid using "left" or "right".*
> *3. Output only the final caption.*

Figure 9: The Textual Template of MMGP.

# E MORE ANALYSES

## E.1 THE TEXTUAL TEMPLATE OF MMGP

As shown in Figure 9, the textual template we designed has a very simple structure and can achieve satisfactory results.

## E.2 MULTI-VIEW CONSISTENCY STUDY

To better demonstrate that FineSplat produces consistent captions across multiple views, we visualized the feature maps from different views, and the results are shown in the Figure 10. From the results, it can be observed that within the same scene, the same object exhibits consistent feature rep-

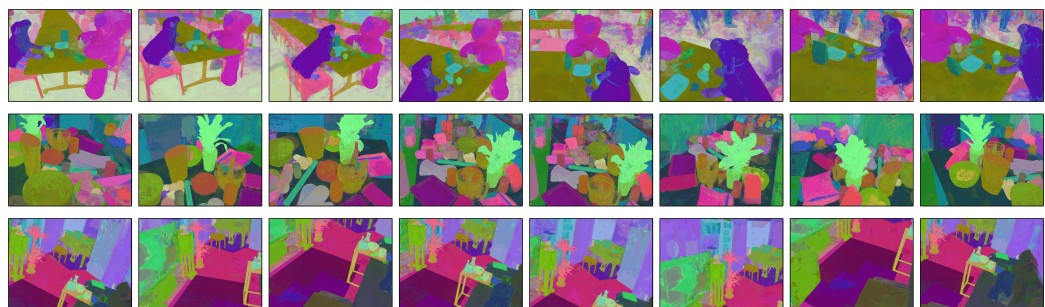

Figure 10: Visualization results of feature maps rendered from different views, where the first row corresponds to the teatime scene, the second row to the counter scene, and the third row to the room scene.

Table 10: Ablation study of the Gaussian language feature dimension on the *Teatime* scene of Fine-OVS.

| $d$ | 3 | 4 | 5 | 6 | 9 | 12 |
|---|---|---|---|---|---|---|
| Acc (%) | 80.4 | 78.4 | 82.4 | **90.2** | **90.2** | 88.3 |
| IoU (%) | 42.7 | 46.0 | 48.1 | 53.1 | **53.6** | 52.9 |

resentations across different views (with similar colors indicating similar features). This indicates that the captions for the same object converge to consistent representations after training, further demonstrating the robustness of FineSplat in maintaining view-consistent semantics.

### E.3  Effectiveness of the Multi-Modal Guided Prompting

To more clearly illustrate the refinement effect of the Multi-Modal Guided Prompting (MMGP) submodule on the initial captions, we further compare the initial captions of the target objects with the captions refined by MMGP, as shown in Figure 11. The results show that while the captions generated by DAM provide relatively detailed descriptions of the target objects, they still lack spatial relationships and functionality, which are crucial for fine-grained scene understanding. After being processed by our proposed MMGP submodule, the refined captions not only compensate for these missing attributes but also eliminate redundant descriptions.

### E.4  More Ablation Study

**Ablation Study of the Gaussian Language Feature Dimension**   To explore the impact of the Gaussian language feature dimension on model performance, we conduct an ablation study on the *Teatime* scene of Fine-OVS, with the results shown in Table 10. The results indicate that as the feature dimension increases, performance first improves and then declines, achieving the best result at $d = 9$. However, considering the trade-off between computational cost and runtime efficiency, we adopt $d = 6$, which offers a balanced compromise between accuracy and efficiency, in all experiments.

**Ablation Study of the Text Encoder**   To validate the rationale of formulating the process as asymmetric text retrieval tasks, specifically, to demonstrate the effectiveness of the text encoder used in the FGFFM, we conduct additional experiments using different combinations of text encoders, and the results are shown in Table 11 and Table 12. From the results, we observe that when the process is reformulated as feature matching (i.e., employing the text-matching encoder during both training and inference), the localization accuracy and segmentation IoU on Fine-OVS decrease by 5.3% and 3.1%, respectively. This further validates the rationality and effectiveness of modeling this process in FineSplat as asymmetric text-retrieval tasks.

| RGB Image | Initial Caption | Refined Caption |
|---|---|---|
| | *"A white paper napkin with a folded corner, displaying a soft texture and a slightly crumpled appearance."* | *"A white paper napkin with a folded corner, soft texture, and slightly crumpled appearance, used for cleaning and placed on the table."* |
| | *"glossy red apple with a smooth surface and a slight gradient of color, ranging from deep red to a lighter red with hints of yellow near the top."* | *"A glossy red apple with a smooth surface is used for eating and sits on the table."* |
| | *"A white book with a vertical orientation, featuring a black spine. The cover displays a partial image of a kitchen scene with a counter and various items."* | *"A white, paper book with a black spine and glossy texture, used for reading, positioned on a shelf."* |
| | *"A knife with a straight, narrow blade that tapers to a pointed tip, featuring a reflective metallic surface. The handle is black with a textured grip and three visible rivets."* | *"The knife has a black, textured grip with three rivets, a reflective metallic blade, and is used for cutting, hanging on the wall."* |
| | *"A vibrant orange with a slightly rough, dimpled texture and a small, darkened spot near the top."* | *"A vibrant orange with a slightly rough, dimpled texture and a small, darkened spot near the top is used for eating and is placed on the countertop."* |
| | *"A rectangular metal baking tray with rounded corners and a textured surface featuring a chevron pattern."* | *"A rectangular metal baking tray with a textured chevron pattern is used for baking and placed on a kitchen countertop."* |

Figure 11: Comparison of initial captions and MMGP-refined captions for target objects. The target objects are highlighted with red bounding boxes in the RGB images.

Table 11: The mean accuracy(%) of 3D object localization on the Fine-OVS using different combinations of text encoders.

| Encoder mode | Fine-OVS | | | | |
|---|---|---|---|---|---|
| | Teatime | Ramen | Kitchen | Figurines | Overall |
| matching-matching | 88.1 | 85.4 | 80.4 | 80.9 | 83.7 |
| query-query | 85.5 | 86.7 | 83.3 | 82.6 | 84.5 |
| passage-passage | 81.2 | 75.9 | 84.2 | 69.1 | 77.6 |
| passage-query | **87.0** | **89.2** | **90.2** | **89.7** | **89.0** |

### E.5  FINE-GRAINED ATTRIBUTE AWARENESS OF FINESPLAT

We present additional 3D object localization visualizations on the Fine-OVS to validate the fine-grained attribute awareness of FineSplat, as shown in Figure 12. Specifically, we select examples that satisfy same category but differing in only one fine-grained attribute and examine FineSplat's localization results for these test queries. The results demonstrate that FineSplat can correctly distinguish objects that differ in only a single fine-grained attribute, indicating that the model truly understands each fine-grained semantic meaning and is an effective solution for fine-grained scene understanding.

### E.6  COMPUTATIONAL COST ANALYSIS OF FINESPLAT

Table 12: The mean IoU(%) of 3D emantic segmentation on the Fine-OVS using different combinations of text encoders.

| Encoder mode | Fine-OVS | | | | |
|---|---|---|---|---|---|
| | Teatime | Ramen | Kitchen | Figurines | Overall |
| matching-matching | 52.1 | 60.2 | 50.3 | 47.3 | 52.5 |
| query-query | 47.2 | 60.7 | 53.4 | 51.7 | 53.3 |
| passage-passage | 41.7 | 55.4 | 48.9 | 42.9 | 47.2 |
| passage-query | **54.2** | **61.6** | **53.1** | **53.4** | **55.6** |

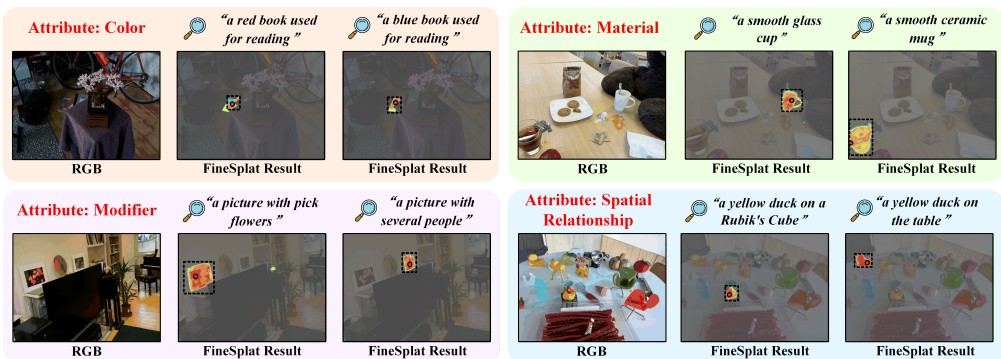

Figure 12: 3D object localization visualizations of FineSplat for fine-grained attribute awareness.

Table 13: The full computational cost (time cost and VRAM cost) analysis for the entire pipeline of FineSplat.

| Method | SAM and DAM / SAM and CLIP | MLLM and Text Encoder | Autoencoder Training | Language Gaussian Training (30k iters) | Total Time / Max VRAM |
|---|---|---|---|---|---|
| FineSplat | 2.6h / 9.8G | 3.8h / 19.5G | 0.1h / 0.5G | 0.4h / 6G | 7.0h / 19.5G |
| LangSplatV2 | 1.2h / 7G | - / - | - / - | 10.5h / 20.9G | 11.7h / 20.9G |

We conduct a full computational cost analysisfor the entire pipeline on a single 24G RTX-4090 GPU, as shown in the Table 13. Compared to LangSplatV2, FineSplat has advantages in both total training time and maximum VRAM cost.

E.7 THE GENERALIZATION ABILITY OF FINESPLAT

To further evaluate the generalization capability of FineSplat, *i.e.*, its performance on out-of-distribution queries, we conducted additional experiments, as shown in Tables 14 and 15. Specifically, before constructing the fine-grained feature field, we removed a certain fine-grained attribute from the captions generated by the FGCG strategy, such as material or functionality. The results show that after removing the material and functionality attributes separately, FineSplat's overall accuracy in 3D object localization only decreased by 8.5% and 5.1%, respectively, while the queries containing material and functionality accounted for 29.2% and 25.4% of all queries. Similarly, in the 3D semantic segmentation task, the overall IoU only decreased by 7.3% and 6.1%. These results indicate that FineSplat maintains strong robustness even when handling out-of-distribution queries, demonstrating its excellent generalization ability.

We conduct a more targeted generalization study to further validate the generalization capability of FineSplat. Specifically, for the material attribute, we select all queries containing material attribute in the corresponding scenes and computed the change in localization accuracy (%) before and after removing this attribute. We perform the same analysis for the function attribute. The results are shown in the Table 16 and Table 17. The results indicate that even after completely removing the material or functionality attributes, FineSplat is still able to correctly localize most queries that rely

Table 14: Quantitative results of 3D object localization on the Fine-OVS after removing a specific attribute from captions generated by the FGCG strategy. We report the mean accuracy(%). "Percentage" denotes the percentage of queries containing the corresponding attribute among all queries.

| Method | Percentage | Fine-OVS | | | | |
|---|---|---|---|---|---|---|
| | | Teatime | Kitchen | Room | Counter | Overall |
| FineSplat | - | **87.0** | **90.2** | **89.4** | **92.9** | **89.9** |
| w/o material | 29.2% | 79.8 △8.2% | 78.5 △12.9% | 83.6 △6.5% | 86.8 △6.5% | 82.2 △8.5% |
| w/o functionality | 25.4% | 84.1 △3.3% | 80.4 △8.6% | 86.4 △3.3% | 90.1 △3.0% | 85.3 △5.1% |

Table 15: Quantitative results of 3D semantic segmentation on the Fine-OVS after removing a specific attribute from captions generated by the FGCG strategy. We report the mean IoU(%). "Percentage" denotes the percentage of queries containing the corresponding attribute among all queries.

| Method | Percentage | Fine-OVS | | | | |
|---|---|---|---|---|---|---|
| | | Teatime | Kitchen | Room | Counter | Overall |
| FineSplat | - | **54.2** | **53.1** | **62.4** | **64.3** | **58.5** |
| w/o material | 29.2% | 49.7 △8.3% | 47.3 △10.9% | 61.2 △1.9% | 58.6 △8.8% | 54.2 △7.3% |
| w/o functionality | 25.4% | 49.8 △8.1% | 49.4 △6.9% | 60.4 △3.2% | 60.1 △6.5% | 54.9 △6.1% |

on these attributes, with only a minor performance drop. This further demonstrates that FineSplat maintains strong generalization ability even when key fine-grained attributes are absent.

### E.8 MORE QUALITATIVE COMPARISONS OF 3D OBJECT LOCALIZATION

As shown in Figure 13, we provide more visualizations of 3D object localization results from different methods. The results indicate that, compared to other methods, our proposed FineSplat can accurately handle a variety of fine-grained queries. For instance, in the *Teatime* scene, given the query "a white tag with DALL·E brand" although GAGS and LangSplatV2 both obtain correct localization results, their activated regions extend far beyond the actual extent of the target object. In contrast, our method achieves accurate localization and produces an activated region that closely matches the target object.

### E.9 MORE QUALITATIVE COMPARISONS OF 3D SEMANTIC SEGMENTATION

To more intuitively demonstrate the superior performance of FineSplat over other methods in open-vocabulary fine-grained scene understanding, we present the 3D semantic segmentation visualization results on the Fine-OVS dataset, as shown in Figures 14 and 15. Given fine-grained queries, the semantic segmentation results of LangSplat, GAGS, and LangSplatV2 are suboptimal. On the one hand, these methods may fail to accurately activate the target region, resulting in incorrect segmentation. On the other hand, even when the target is successfully activated, they often concurrently activate additional regions, leading to accurate 3D object localization but significantly lower 3D semantic segmentation IoU scores. In contrast, our proposed FineSplat not only accurately activates the target object regions, but also produces activation boundaries that closely align with the Ground Truth, significantly outperforming existing methods. The results demonstrate that FineSplat achieves both high precision and high consistency, providing an effective solution for fine-grained scene understanding.

## F LIMITATION

Although the proposed FineSplat demonstrates remarkable performance in the fine-grained scene understanding task, it still has several limitations that require further improvement. Specifically, FineSplat builds on pretrained foundation models, which not only increase GPU memory requirements but also result in longer computational overhead. In addition, training an autoencoder for each scene to compress features, while improving efficiency, inevitably leads to information loss

Table 16: Quantitative results of 3D object localization on the Fine-OVS, before and after removing the material attribute.

| Method | Fine-OVS | | | | |
| --- | --- | --- | --- | --- | --- |
| | Teatime | Kitchen | Room | Counter | Overall |
| w/ material | 83.3% (15/18) | 92.0% (23/25) | 100% (11/11) | 100% (25/25) | 93.8% |
| w/o material | 77.8% (14/18) | 84.0% (21/25) | 90.9% (10/11) | 96.0% (24/25) | 87.2% |

Table 17: Quantitative results of 3D object localization on the Fine-OVS, before and after removing the functionality attribute.

| Method | Fine-OVS | | | | |
| --- | --- | --- | --- | --- | --- |
| | Teatime | Kitchen | Room | Counter | Overall |
| w/o functionality | 85.0% (17/20) | 100% (11/11) | 93.3% (14/15) | 95.0% (19/20) | 93.3% |
| w/o functionality | 80.0% (16/20) | 81.8% (9/11) | 80.0% (12/15) | 90.0% (18/20) | 83.0% |

and lacks generalization to other scenes. Therefore, future work could focus on modeling cross-scene consistency to further enhance both the effectiveness and generalization of fine-grained scene understanding.

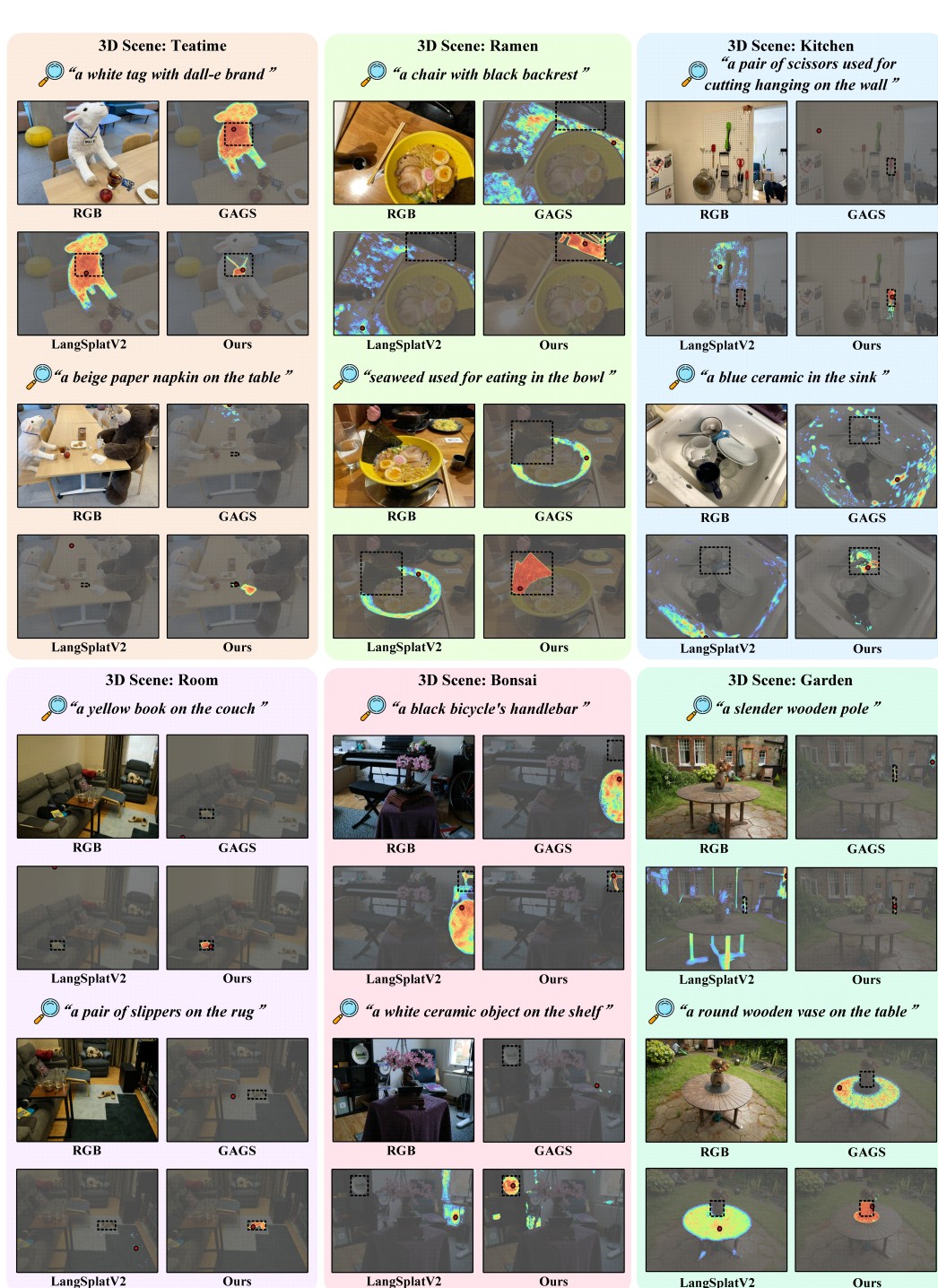

Figure 13: More qualitative comparisons of fine-grained open-vocabulary 3D object localization on the Fine-OVS dataset. Black dashed boxes denote the annotated regions, and red points correspond to the predictions of the model.

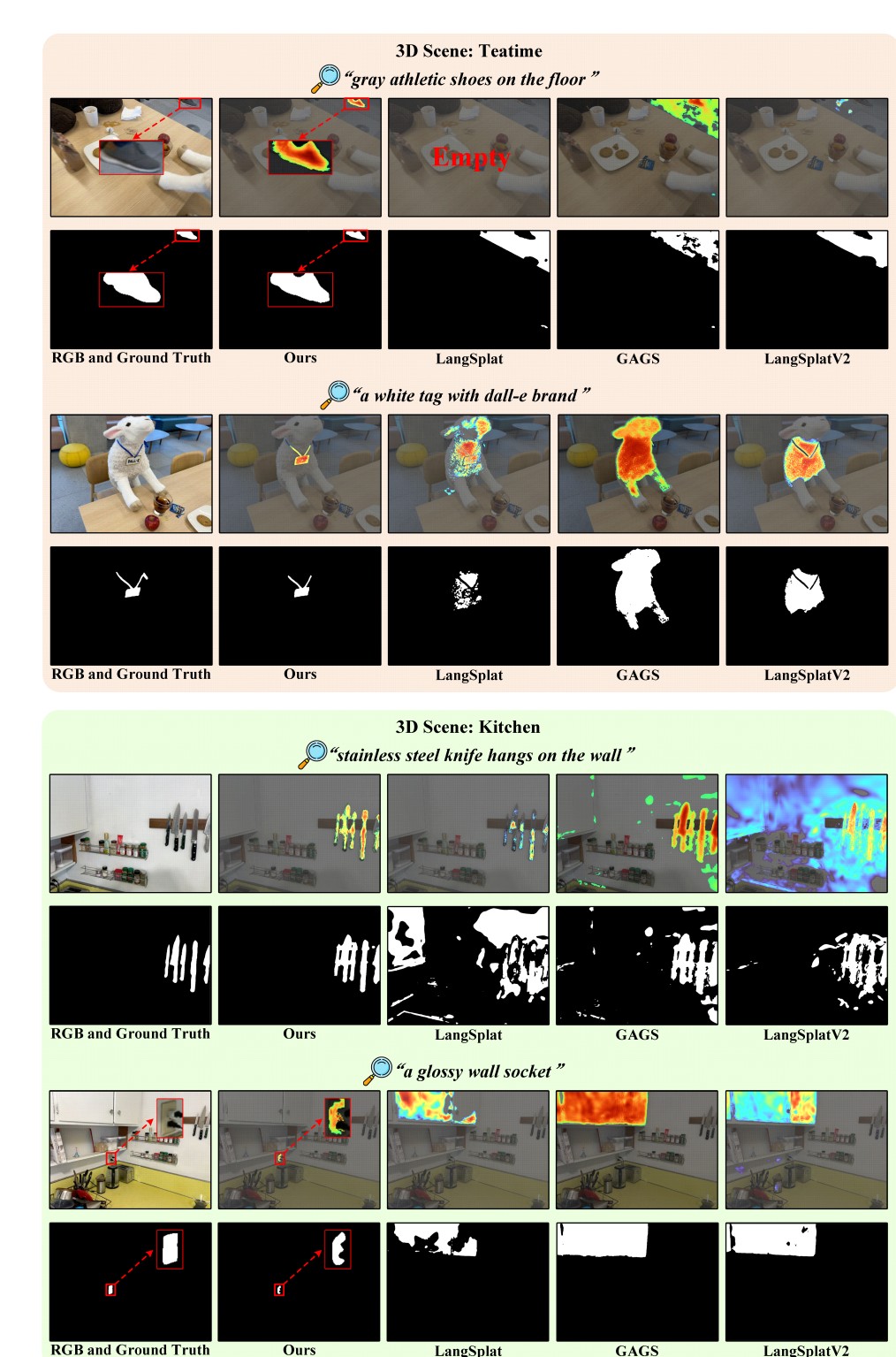

Figure 14: More qualitative comparisons of fine-grained open-vocabulary 3D semantic segmentation on the Fine-OVS dataset.

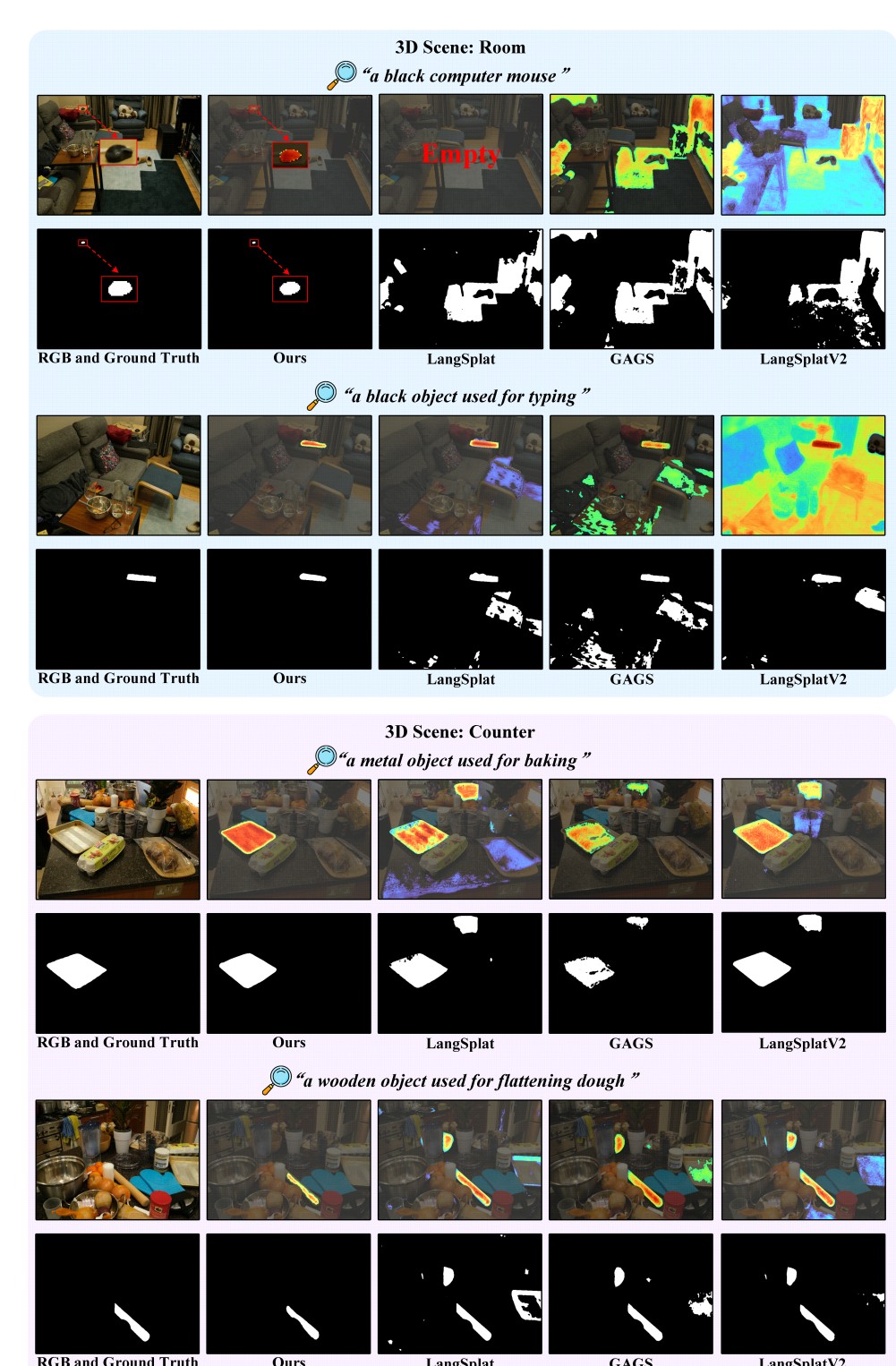

Figure 15: More qualitative comparisons of fine-grained open-vocabulary 3D semantic segmentation on the Fine-OVS dataset.

