# OpenReview forum: "FineSplat: Fine-Grained 3D Open-Vocabulary Language Gaussian Splatting"
_ICLR.cc/2026/Conference — Submitted to ICLR 2026_

### Official Review · Reviewer_AMGd · 2025-10-29

**Soundness:** 3
**Presentation:** 4
**Contribution:** 3
**Rating:** 8
**Confidence:** 5

**Summary:**

This paper introduces a challenging task of fine-grained open-vocabulary scene understanding, which is of great significance for real-world interaction. A novel language gaussian splatting framework, FineSplat, is proposed to address this task. This paper also introduces a novel benchmark dataset, Fine-OVS, to address the gap in previous benchmarks where the test queries are limited to simple category-level labels. The authors conducted extensive qualitative and quantitative experiments to verify the effectiveness of the proposed method from multiple perspectives. Overall, this is an interesting paper that makes a meaningful contribution to the community.

**Strengths:**

1.The paper is well written and well organized, and the proposed FineSplat is easy to follow.
2.This paper has a strong motivation and breaks through the CLIP-based paradigm by modeling the feature field using only fine-grained captions, which is a novel idea.
3.To support research and evaluation on fine-grained scene understanding tasks, this paper constructs a novel benchmark, Fine-OVS, which includes 8 fine-grained attributes.
4.Extensive quantitative and qualitative experimental results demonstrate that FineSplat exhibits stronger fine-grained understanding capabilities compared to baseline methods.

**Weaknesses:**

1.The paper lacks ablation studies on Fine-Grained Feature Field Modeling. Specifically, if this process is treated as feature matching rather than text retrieval, how the choice of encoder affects performance remains unclear.
2.In Figure 2, the two text encoders are shown using the same color. However, according to the paper’s description, they are two different encoders, so the visual representation in Figure 2 should be adjusted.
3.The authors should clarify whether they plan to release the benchmark publicly. I believe this benchmark could significantly advance research in fine-grained scene understanding tasks.

**Questions:**

Please see the weaknesses.

---

> ### Author Response · Authors · 2025-11-19
> **Response to Reviewer AMGd**
>
> We sincerely appreciate you for your careful review, insightful comments, and recognition of **the paper writing, strong motivation, novel idea, novel benchmark, and SOTA performance**. We tried our best to povide detailed clarifications for all your concerns as follows. Please do not hesitate to share any additional questions. We would be delighted to answer further.
>
> > **(Weakness.1)** The paper lacks ablation studies on Fine-Grained Feature Field Modeling. Specifically, if this process is treated as feature matching rather than text retrieval, how the choice of encoder affects performance remains unclear.
>
> **A:** Thank you for your valuable suggestion! We have added ablation studies for the Fine-Grained Feature Field Modeling strategy. Specifically, we conducted additional experiments using different combinations of text encoders, and the results are shown in tables below.
>
> The mean accuracy(%) of 3D object localization on the Fine-OVS:
>
> | Encoder mode       | **Teatime** | **Ramen** | **Kitchen** | **Figurines** | **Overall** |
> |--------------------|-------------|-----------|-------------|----------------|-------------|
> | matching-matching  | 88.1        | 85.4      | 80.4        | 80.9           | 83.7        |
> | query-query        | 85.5        | 86.7      | 83.3        | 82.6           | 84.5        |
> | passage-passage    | 81.2        | 75.9      | 84.2        | 69.1           | 77.6        |
> | passage-query      | **87.0**    | **89.2**  | **90.2**    | **89.7**       | **89.0**    |
>
> The mean IoU(%) of 3D semantic segmentation on the Fine-OVS:
>
> | Encoder mode       | **Teatime** | **Ramen** | **Kitchen** | **Figurines** | **Overall** |
> |--------------------|-------------|-----------|-------------|----------------|-------------|
> | matching-matching  | 52.1        | 60.2      | 50.3        | 47.3           | 52.5        |
> | query-query        | 47.2        | 60.7      | 53.4        | 51.7           | 53.3        |
> | passage-passage    | 41.7        | 55.4      | 48.9        | 42.9           | 47.2        |
> | passage-query      | **54.2**    | **61.6**  | **53.1**    | **53.4**       | **55.6**    |
>
>
> From the results, we observe that when the process is reformulated as **feature matching** (i.e., **employing the text-matching encoder during both training and inference**), the localization accuracy and segmentation IoU on Fine-OVS decrease by 5.3% and 3.1%, respectively. This further validates the rationality and effectiveness of modeling this process in FineSplat as asymmetric text-retrieval tasks.
>
> In addition, we conducted additional ablation studies on the query encoder and passage encoder, as shown in the second and third rows of tables. The results show that both text-encoder selection strategies lead to a consistent performance degradation. All of the above ablation studies have been incorporated into the revised paper **(Appendix E.4)**.
>
> > **(Weakness.2)** In Figure 2, the two text encoders are shown using the same color. However, according to the paper’s description, they are two different encoders, so the visual representation in Figure 2 should be adjusted.
>
> **A:** Thanks for your careful review and valuable suggestion. Indeed, the two encoders used in the training and testing stages are different. In the revised paper, we have revised Figure 2 to clearly distinguish the two encoders using different colors, so as to more accurately correspond to the descriptions in the paper.
>
> > **(Weakness.3)** The authors should clarify whether they plan to release the benchmark publicly. I believe this benchmark could significantly advance research in fine-grained scene understanding tasks.
>
> **A:** Thanks for your recognition of the contribution of our constructed benchmark. We plan to release the Fine-OVS benchmark in the future, including all annotated queries for all scenes, and will also make the corresponding code publicly available, with the aim of advancing research on fine-grained scene understanding tasks. In future work, we also plan to further expand the scale of Fine-OVS.

---

> > ### Comment · Reviewer_AMGd · 2025-11-22
> > **Response to the Rebuttal.**
> >
> > Thanks for the authors' detailed response. Framing the problem as an "asymmetric text retrieval" task is an inspiring idea, and the additional experiments provided by the authors further validate its effectiveness. The author's response also solved most of my problems and concerns. Overall, I will keep the original score and recommend acceptance as poster.

---

> > > ### Author Response · Authors · 2025-11-24
> > > **Official Comment by Authors**
> > >
> > > Dear reviewer AMGd,
> > >
> > > We are glad that our responses could solve your concerns. Thank you again for taking the time to review our paper and providing the insightful comments!
> > >
> > > Best regards,
> > >
> > > Authors

---

### Official Review · Reviewer_uodS · 2025-10-30

**Soundness:** 3
**Presentation:** 3
**Contribution:** 3
**Rating:** 4
**Confidence:** 3

**Summary:**

This paper introduces a method for semantic 3D feature fields. Authors observes that CLIP often behaves like a bag-of-words under phrase/sentence-level queries and fails to capture rich compositional context. To address this, they propose an embedded 3DGS that ingests long, descriptive text features. The main idea is introduction of coarse-to-fine captioning pipeline that leverages a DAM and an MLLM to extract fine-grained textual signals, and they adjust the text encoder to handle unbalanced captions (mismatched information density between user queries and generated captions). To evaluate fine-caption-conditioned querying, they curate a new dataset for 3D localization from fine captions and show improvements over prior methods on tasks requiring detailed textual understanding.

**Strengths:**

- Clear motivation : Identification of CLIP’s bag-of-words-like behavior under long queries and the proposal to inject richer text signals into 3DGS. The problem framing (descriptive queries vs. keyword queries) is persuasive and grounded in practical use cases.

- The captioning pipeline and dataset can catalyze follow-up research on fine-grained 3D language grounding.

**Weaknesses:**

**More Ablation experiments?**
- It is unclear whether the method truly understands each fine-grained meaning; ablations are needed (e.g., Observation on if they can distinguish same category with different materials, or when spatial relations are identifiable).

**Multi-view consistency**
- Multi-view inconsistency remains a concern for text signals.
Prior works note instability from view-dependent cues due to inconsistent SAM segmentation and occlusions.
Suggest text multi-view bootstrapping experiments: does aggregating captions from different views converge to consistent 3D semantics?

**Compatible with existing dataset**
- Do results on existing datasets remain comparable? While expressiveness seems improved, it is unclear whether coarse information is still well represented.

**About generalization study**
- In the generalization study (Appendix E.4), comparisons should focus only on changed factors; since material accounts for only 29.2%, small overall changes may not indicate true generalization. How does performance differs using changing cases only.

**Questions:**

- Is there an ablation without the unbalanced-query/caption mechanism?
- Other questions are covered under Weaknesses above.

---

> ### Author Response · Authors · 2025-11-19
> **Response to Reviewer uodS (Part 1/2)**
>
> We sincerely appreciate you for your careful review, insightful comments, and recognition of the **clear motivation, novel pipeline, and novel benchmark**. We tried our best to povide detailed clarifications for all your concerns as follows. Please do not hesitate to share any additional questions. We would be delighted to answer further.
>
> > **(Weakness.1)** More Ablation experiments? It is unclear whether the method truly understands each fine-grained meaning; ablations are needed (e.g., Observation on if they can distinguish same category with different materials, or when spatial relations are identifiable)
>
> **A:** Thanks for your valuable suggestion! We conducted additional ablation studies to evaluate the fine-grained attribute awareness of FineSplat. Due to the limited number of instances in scenes that satisfy the condition of **same category but differing in only one fine-grained attribute**, we systematically screened the entire benchmark and listed valid test queries as follows:
> * **Color Attribute：** “a red book used for reading” and “a blue book used for reading”.
> * **Material Attribute:** “a smooth glass cup” and “a smooth ceramic mug”.
> * **Modifier Attribute:** “a picture with pink flowers” and “a picture with several people”.
> * **Spatial Relation Attribute:** “a yellow duck on a Rubik's Cube” and “a yellow duck on the table”.
>
> Given these queries, the 3D object localization visualizations of FineSplat are provided in **Appendix E.5** of the revised paper. The results demonstrate that **FineSplat can correctly distinguish objects that differ in only a single fine-grained attribute**, indicating that the model truly understands each fine-grained semantic meaning and is an effective solution for fine-grained scene understanding.
>
> > **(Weakness.2)** Multi-view consistency. Multi-view inconsistency remains a concern for text signals. Prior works note instability from view-dependent cues due to inconsistent SAM segmentation and occlusions. Suggest text multi-view bootstrapping experiments: does aggregating captions from different views converge to consistent 3D semantics?
>
> **A:** Thanks for your valuable comments and suggestion. SAM may indeed produce inconsistent segmentations of the same object across different views when occlusions occur. To alleviate this, we generate masks using multi semantic levels of SAM, obtaining both **whole-level and part-level masks**, which improves the cross-view stability of the segmentation.
>
> In addition, to verify whether the generated captions exhibit multi-view consistency, **we visualize the feature maps rendered from different views of the corresponding scene**. The results are presented in the **Appendix E.2** of the revised paper. From the results, we can observe that within the same scene, **the same object exhibits consistent feature representations across different views (with similar colors indicating similar features**). This indicates that the captions for the same object converge to consistent representations after training, **further demonstrating the robustness of FineSplat in maintaining view-consistent semantics**.
> > **(Weakness.3)** Compatible with existing dataset. Do results on existing datasets remain comparable? While expressiveness seems improved, it is unclear whether coarse information is still well represented.
>
> **A:** Thanks for your comments. **Yes, FineSplat remains compatible with existing coarse-grained datasets.** We have provided in **Appendix D.2** the results of FineSplat, which was designed specifically for fine-grained scene understanding, on the LERF dataset. In addition, we further evaluate the model after making slight adjustments (modifying the text encoder) **on the general scene understanding task** that includes both coarse-grained and fine-grained queries.
>
> The experimental results of 3D object localization on the Fine-OVS (fine-grained):
> | Method       | Teatime | Ramen | Kitchen | Figurines | Overall |
> |--------------|---------|-------|---------|-----------|---------|
> | LangSplat_V2 | 69.5    | 77.1  | 49.0    | 70.6      | 66.6    |
> | Ours         | 88.1    | 85.4  | 80.4    | 80.9      | **83.7** |
>
> The experimental results of 3D object localization on the LERF (coarse-grained):
> | Method       | Teatime | Ramen | Kitchen | Figurines | Overall |
> |--------------|---------|-------|---------|-----------|---------|
> | LangSplat_V2 | 88.1    | 71.8  | 68.2    | 82.1      | 77.6    |
> | Ours         | 96.6    | 69.2  | 66.8    | 78.6      | **77.8** |
>
> The results show that FineSplat maintains overall superior performance on LERF, demonstrating strong **coarse-to-fine consistency**. The method not only preserves robust representational ability for coarse-grained categories, but also successfully models more complex fine-grained feature fields.

---

> ### Author Response · Authors · 2025-11-19
> **Response to Reviewer uodS (Part 2/2)**
>
> > **(Weakness.4)** About generalization study. In the generalization study (Appendix E.4), comparisons should focus only on changed factors; since material accounts for only 29.2%, small overall changes may not indicate true generalization. How does performance differs using changing cases only.
>
>
> **A:** Thank you for your valuable suggestion! Following your suggestion, we conducted a more targeted generalization study to further validate the generalization capability of FineSplat. Specifically, for the material attribute, we selected all queries containing material attribute in the corresponding scenes and **computed the change in localization accuracy (%) before and after removing this attribute**. We performed the same analysis for the function attribute. The results are shown in the tables below:
>
> | Method                 | Teatime        | Kitchen         | Room            | Counter          | Overall  |
> |------------------------|----------------|------------------|------------------|-------------------|----------|
> | FineSplat w/ material  | 83.3% (15/18)  | 92.0% (23/25)    | 100% (11/11)     | 100% (25/25)      | 93.8%   |
> | FineSplat w/o material | 77.8% (14/18)  | 84.0% (21/25)    | 90.9% (10/11)    | 96.0% (24/25)     | 87.2%   |
>
> | Method                       | Teatime        | Kitchen        | Room           | Counter        | Overall |
> |------------------------------|----------------|-----------------|----------------|----------------|---------|
> | FineSplat w/ functionality  | 85.0% (17/20)  | 100% (11/11)    | 93.3% (14/15)  | 95.0% (19/20)  | 93.3%  |
> | FineSplat w/o functionality  | 80.0% (16/20)  | 81.8% (9/11)    | 80.0% (12/15)  | 90.0% (18/20)  | 83.0%  |
>
> The results indicate that even after completely removing the material or functionality attributes, FineSplat is still able to correctly localize most queries that rely on these attributes, with only a minor performance drop. **This further demonstrates that FineSplat maintains strong generalization ability even when key fine-grained attributes are absent.** We have added the above-mentioned experiments and analyses to Appendix E.7.
> > **(Question.1)** Is there an ablation without the unbalanced-query/caption mechanism?
>
> **A:** Thanks for your suggestion. We have added experiments related to the unbalanced-query/caption mechanism. Specifically, in fine-grained scene understanding tasks, the queries in the inference stage often contain only a subset of attributes, making them unbalanced compared to the captions generated during training. Therefore, we formulate the task as **asymmetric text retrieval**, using a passage encoder during training and a query encoder during inference. To further examine the impact of the balanced-query/caption mechanism on performance, we reformulate the task as feature matching, replacing the text encoders in both training and inference with matching encoders, passage encoders, and query encoder, the results are shown in tables below.
>
>   The mean accuracy(%) of 3D object localization on the Fine-OVS:
>
>   | Encoder mode       | **Teatime** | **Ramen** | **Kitchen** | **Figurines** | **Overall** |
>   |--------------------|-------------|-----------|-------------|----------------|-------------|
>   | matching-matching  | 88.1        | 85.4      | 80.4        | 80.9           | 83.7        |
>   | query-query        | 85.5        | 86.7      | 83.3        | 82.6           | 84.5        |
>   | passage-passage    | 81.2        | 75.9      | 84.2        | 69.1           | 77.6        |
>   | passage-query      | **87.0**    | **89.2**  | **90.2**    | **89.7**       | **89.0**    |
>
>   The mean IoU(%) of 3D semantic segmentation on the Fine-OVS:
>
>   | Encoder mode       | **Teatime** | **Ramen** | **Kitchen** | **Figurines** | **Overall** |
>   |--------------------|-------------|-----------|-------------|----------------|-------------|
>   | matching-matching  | 52.1        | 60.2      | 50.3        | 47.3           | 52.5        |
>   | query-query        | 47.2        | 60.7      | 53.4        | 51.7           | 53.3        |
>   | passage-passage    | 41.7        | 55.4      | 48.9        | 42.9           | 47.2        |
>   | passage-query      | **54.2**    | **61.6**  | **53.1**    | **53.4**       | **55.6**    |
>
>   The results show that all three balanced-query/caption mechanisms (feature matching) exhibit some performance degradation. In contrast, the passage-query encoder corresponding to the unbalanced-query/caption mechanism (asymmetric text retrieval) designed in this work achieves consistently superior performance, demonstrating the rationality and effectiveness of this mechanism. We have added the above-mentioned experiments and analyses to **Appendix E.4**.

---

> ### Author Response · Authors · 2025-11-27
> **Looking Forward to Discussion**
>
> Dear Reviewer uodS,
>
> We sincerely appreciate the time and effort you have dedicated to reviewing our paper.
>
> As the discussion period is drawing to a close, we would like to kindly check whether you have any remaining questions or points you would like us to clarify. We would be glad to provide any further information and are very grateful for any additional feedback you may have.
>
> Thank you very much, and we look forward to hearing from you!
>
> Best regards,
>
> Authors

---

### Official Review · Reviewer_v764 · 2025-10-30

**Soundness:** 2
**Presentation:** 3
**Contribution:** 2
**Rating:** 4
**Confidence:** 4

**Summary:**

The authors identify a limitation in current open-vocabulary 3D understanding methods, which are mostly limited to coarse, object-level recognition. They propose a new task called "Fine-grained Open-vocabulary 3D Scene Understanding" and a framework, FineSplat, to address it. The core idea of FineSplat is to avoid using vision-language models like CLIP for feature matching. Instead, it generates detailed textual captions for all objects in a scene and models the 3D feature field using only text-based embeddings. This reframes the problem from cross-modal (vision-text) matching to intra-modal (text-text) retrieval. The method involves a Fine-Grained Caption Generation (FGCG) strategy and a Fine-Grained Feature Field Modeling (FGFFM) strategy. The authors also introduce a new small-scale benchmark, Fine-OVS, to evaluate this fine-grained task

**Strengths:**

1. **Excellent Motivation:** The paper is well-motivated. It correctly identifies a key weakness in current 3D language-field methods (e.g., LangSplat, GAGS) that rely on CLIP: they struggle with fine-grained attribute binding and compositionality, often behaving like a "bag-of-words" . The examples in Figure 1 clearly illustrate this failure mode.
2. **Novel Problem Formulation:** The core idea of reformulating the task from vision-text matching to text-text retrieval is clever. This directly sidesteps the known attribute-binding issues of models like CLIP and grounds the scene representation in a semantically richer (text-only) space.
3. **New Benchmark Contribution:** The paper introduces Fine-OVS, a new benchmark specifically designed to evaluate this more challenging fine-grained task, which is a valuable contribution to the community.

**Weaknesses:**

1. **Extremely Complex Pipeline:** The primary weakness of this paper is the immense complexity of the proposed pipeline, which feels more like a heavy engineering effort than a clean, scalable method. The "Fine-Grained Caption Generation" (FGCG) strategy alone (see Fig. 2) requires running multiple, large foundation models in sequence: (1) Run SAM to get masks ; (2) Run DAM (a caption model) on *every* mask ; (3) Run an MLLM (Qwen-VL) with complex multi-modal prompts (including blurred and highlighted images) to *refine* the captions. This multi-stage, computationally massive data-generation process seems impractical to scale.
2. **Scene-Specific Components:** The method's scalability is severely limited by the fact that it requires training a *new, scene-specific* autoencoder for *every single scene*. This is also mentioned as a limitation by the authors. This means FineSplat is not a generalizable, "train-once" model, but rather a pipeline that must be partially re-trained for any new scene it encounters, which is a significant drawback.
3. **Very Limited Evaluation:** The new Fine-OVS benchmark is extremely small, consisting of only **8 scenes**. While the method shows strong performance on this custom-built benchmark, this is not a comprehensive evaluation. It is unclear if this complex pipeline is feasible or effective on larger-scale datasets.
4. **Poor Generalization to Standard Tasks:** The authors' own experiments on the standard (coarse-grained) LERF benchmark (Table 8) show that FineSplat performs *worse* than the baseline LangSplatV2 . This strongly suggests that the method has been over-specialized for its own narrow, fine-grained task and has lost the ability to perform well on general, coarse-grained queries. This supports the idea that this is a "niche" solution that does not advance general scene understanding.

**Questions:**

1. Could the authors provide a full computational cost analysis for the *entire* pipeline? This should include the FGCG data generation (cost of running SAM, DAM, and MLLM on all views) and the FGFFM (cost of training the *per-scene*autoencoder). How many hours/VRAM does it take to process one scene from start to finish, compared to LangSplatV2?
2. The poor performance on LERF (Table 8) is concerning. Does this imply a fundamental trade-off, where the model gains fine-grained accuracy by sacrificing coarse-grained accuracy? Can the model no longer reliably answer simple queries like "find the mug"?

---

> ### Author Response · Authors · 2025-11-19
> **Response to Reviewer v764 (Part 1/3)**
>
> We sincerely appreciate you for your careful review, insightful comments, and recognition of our work’s **excellent motivation, novel method, and benchmark contribution**. We tried our best to povide detailed clarifications for all your concerns as follows. Please do not hesitate to share any additional questions. We would be delighted to answer further.
>
> > **(Question.1)**
> > Could the authors provide a full computational cost analysis for the entire pipeline? This should include the FGCG data generation (cost of running SAM, DAM, and MLLM on all views) and the FGFFM (cost of training the per-sceneautoencoder). How many hours/VRAM does it take to process one scene from start to finish, compared to LangSplatV2?
>
> > **(Weakness.1)** Extremely Complex Pipeline: The primary weakness of this paper is the immense complexity of the proposed pipeline, which feels more like a heavy engineering effort than a clean, scalable method. The "Fine-Grained Caption Generation" (FGCG) strategy alone (see Fig. 2) requires running multiple, large foundation models in sequence: (1) Run SAM to get masks ; (2) Run DAM (a caption model) on every mask ; (3) Run an MLLM (Qwen-VL) with complex multi-modal prompts (including blurred and highlighted images) to refine the captions. This multi-stage, computationally massive data-generation process seems impractical to scale.
>
> **A:** Thanks for your insightful suggestion. We conducted a **full computational cost analysis** for the entire pipeline on a single 24G RTX-4090 GPU, as shown in the table below.
> | Method       | SAM and DAM （FineSplat）/ SAM and CLIP (LangSplatV2)    | MLLM and Text Encoder | Autoencoder Training | Language Gaussian Training (30k iters) | Total Time / Max VRAM |
> |--------------|------------------------|---------------------------|-------------------|---------------------------------------|----------------------|
> | FineSplat    | 2.6h / 9.8G            | 3.8h / 19.5G              | 0.1h / 0.5G       | 0.4h / 6G                             | 7.0h / 19.5G         |
> | LangSplatV2  | 1.2h / 7G               | - / -                | - / -             | 10.5h / 20.9G                         | 11.7h / 20.9G        |
>
>
> **Compared to LangSplatV2, FineSplat has advantages in both total training time and maximum VRAM cost**. The language-Gaussian training time of LangSplatV2 is significantly longer than that of FineSplat. This is because, during its Language Gaussian Training stage, LangSplatV2 embeds 512-dimensional CLIP features into Gaussians, leading to high time and VRAM consumption. In contrast, FineSplat trains an autoencoder to reduce the caption-feature dimensionality to 6, substantially lowering both training time and VRAM cost.
>
> In fact, even when using only SAM and DAM in the Feature Encoding stage, FineSplat still achieves state-of-the-art performance in fine-grained scene understanding, without necessarily requiring an MLLM. The results of 3D object localization:
> | Method        | Total Time / Max VRAM |Teatime | Ramen | Kitchen | Figurines | Overall |
> |---------------|---------|---------|-------|---------|-----------|---------|
> | LangSplatV2| 11.7h / 20.9G  | 69.5    | 77.1  | 49.0    | 70.6      | 66.6    |
> | FineSplat (w/o MLLM)|   3.2h / 9.8G| 78.2 | 80.5 | 62.7 | 81.2 | 75.7 |
>
>
> Besides, it is worth noting that **the FGCG and FGFFM strategies are only applied during the training stage and are not required at all during inference**.
>
> In conclusion, **our training pipeline remains efficient in both time and VRAM compared to LangSplatV2, even with its multi-stage design, and is therefore practical to scale to new scenes**. We have add all above experiemnts and analysis in the revised paper (Appendix E.6).

---

> ### Author Response · Authors · 2025-11-19
> **Response to Reviewer v764 (Part 2/3)**
>
> > **(Weakness.2)** Scene-Specific Components: The method's scalability is severely limited by the fact that it requires training a new, scene-specific autoencoder for every single scene. This is also mentioned as a limitation by the authors. This means FineSplat is not a generalizable, "train-once" model, but rather a pipeline that must be partially re-trained for any new scene it encounters, which is a significant drawback.
>
> **A:** Thanks for your comments. **The need to “train for every single scene” originates from the nature of current 3D Gaussian Splatting (3D-GS) models**, which are mostly effective in a per-scene manner (e.g., **LangSplat, LangSplatV2, GAGS, GS-Grouping, all models need to train per scene**). Therefore, **this limitation does not come from our scene-specific autoencoder**; in fact, the autoencoder in FineSplat can be easily trained across scenes.
>
> Yet, due to the per-scene characteristic of 3D GS, we adopt **a lightweight (1.4M parameters) autoencoder** to perform fine-grained scene understanding task, which has demonstrated effective performance. We fully agree **cross-scene understanding based on 3D GS is a very valuable research theme for future exploration**.
>
> > **(Weakness.3)** Very Limited Evaluation: The new Fine-OVS benchmark is extremely small, consisting of only 8 scenes. While the method shows strong performance on this custom-built benchmark, this is not a comprehensive evaluation. It is unclear if this complex pipeline is feasible or effective on larger-scale datasets.
>
>
> **A:** Thanks for your comments. We have listed **statistics of the Fine-OVS in Table 5 in the original paper**. Here is a reference:
>
> | Dataset       | Scene | Test Frame | Attribute | 1 attribute | 2 attributes | 3 attributes | 4 attributes | Total query |
> |---------------|-------|------------|-----------|-------------|---------------|---------------|---------------|--------------|
> | LERF          | 4     | 22         | 3         | 157         | 52            | 0             | 0             | 209          |
> | Mip_Nerf360   | 4     | 17         | 4         | 40          | 89            | 4             | 0             | 133          |
> | Fine-OVS      | 8     | 43         | 8         | 0           | 170           | 268           | 62            | 500          |
>
> Compared with the LERF and Mip_Nerf360 datasets designed for coarse-grained 3D scene understanding, Fine-OVS exhibits **significant improvements in multiple aspects**, including the number of scenes, the diversity of attributes, the scale of queries, and, most importantly, the complexity of the queries.
>
> Moreover, the cost of collecting scene data is very high, but the **data construction pipeline designed in this paper is easy to scale**. In addition, we also verify that FineSplat performs well under **sparse-view settings (Table 3 in the paper)**, which provides a comprehensive demonstration of FineSplat’s strong fine-grained understanding capability. In summary, the Fine-OVS benchmark can comprehensively and effectively evaluate the performance of different models on the fine-grained scene understanding task.

---

> ### Author Response · Authors · 2025-11-19
> **Response to Reviewer v764 (Part 3/3)**
>
> > **(Question.2)** The poor performance on LERF (Table 8) is concerning. Does this imply a fundamental trade-off, where the model gains fine-grained accuracy by sacrificing coarse-grained accuracy? Can the model no longer reliably answer simple queries like "find the mug"?
>
> > **(Weakness.4)** Poor Generalization to Standard Tasks: The authors' own experiments on the standard (coarse-grained) LERF benchmark (Table 8) show that FineSplat performs worse than the baseline LangSplatV2 . This strongly suggests that the method has been over-specialized for its own narrow, fine-grained task and has lost the ability to perform well on general, coarse-grained queries. This supports the idea that this is a "niche" solution that does not advance general scene understanding.
>
>
> *1. Does this imply a fundamental trade-off, where the model gains fine-grained accuracy by sacrificing coarse-grained accuracy?*
>
> **A:** We agree with this point. **Coarse-grained and fine-grained tasks are two contrasting levels classic problems in computer vision**. The proposed FineSplat is designed for fine-grained scene understanding tasks, and achieves 90.1% accuracy and 57.0% mIoU in object localization and semantic segmentation, **surpassing prior arts by 21.6% and 10.3%**, respectively. Although FineSplat does not achieve state-of-the-art performance on some scenes in the coarse-grained LERF dataset, it still delivers comparable results. Besides, **our framework can achieve a balanced performance between fine-grained and coarse-grained queries through subtle adjustments**. See the answer to the next question for details.
>
>
> *2. Can the model no longer reliably answer simple queries like "find the mug"?*
>
> **A:** No, the model can still reliably answer simple category queries, this can be mitigated with **slight modification**. Specifically, this only requires replacing the passage encoder used during training and the query encoder used during inference with a unified text matching encoder. The experimental results are shown in tables below.
>
> The experimental results of 3D object localization on the Fine-OVS (fine-grained):
> | Method       | Teatime | Ramen | Kitchen | Figurines | Overall |
> |--------------|---------|-------|---------|-----------|---------|
> | LangSplat_V2 | 69.5    | 77.1  | 49.0    | 70.6      | 66.6    |
> | Ours         | 88.1    | 85.4  | 80.4    | 80.9      | **83.7** |
>
> The experimental results of 3D object localization on the LERF (coarse-grained):
> | Method       | Teatime | Ramen | Kitchen | Figurines | Overall |
> |--------------|---------|-------|---------|-----------|---------|
> | LangSplat_V2 | 88.1    | 71.8  | 68.2    | 82.1      | 77.6    |
> | Ours         | 96.6    | 69.2  | 66.8    | 78.6      | **77.8** |
>
> The results show that FineSplat achieves the **improved overall performance on both fine-grained and coarse-grained tasks**. Therefore, FineSplat not only retains the fundamental coarse-grained query capabilities but also **successfully models more complex fine-grained feature fields**.
>
>
> *3. This is a "niche" solution that does not advance general scene understanding.*
>
> **A:** From the above results, we demonstrate that FineSplat is flexible and **capable of achieving balanced and advanced performance in both coarse-grained and fine-grained 3D scene understanding**. Therefore, FineSplat is not a "niche" solution; rather, we believe it can significantly advance general scene understanding—especially through **its addressing of a critical missing capability in current methods: fine-grained understanding, which is essential for real-world interaction**.

---

> ### Author Response · Authors · 2025-11-27
> **Looking Forward to Discussion**
>
> Dear Reviewer v764,
>
> We sincerely appreciate the time and effort you have dedicated to reviewing our paper.
>
> As the discussion period is drawing to a close, we would like to kindly check whether you have any remaining questions or points you would like us to clarify. We would be glad to provide any further information and are very grateful for any additional feedback you may have.
>
> Thank you very much, and we look forward to hearing from you!
>
> Best regards,
>
> Authors

---

### Author Response · Authors · 2025-12-01
**Summary of Rebuttal**

Dear AC,

We sincerely appreciate your time and effort throughout the review process for our submission. We thank all the reviewers for their detailed and valuable comments on our work. We highlight the main strengths and contributions of our work as follows:
* **Clear and Excellent Motivation (Reviewer v764,  uodS, and AMGd).**  Existing methods are fundamentally limited to a **category-level coarse-grained understanding**, we argue that real-world interaction demands a more sophisticated capability, which we define as the task of **Fine-grained Open-vocabulary 3D Scene Understanding**.
* **Novel and Reasonable Method (Reviewer v764,  uodS, and AMGd).** We propose the novel FineSplat, which reformulates the problem as a robust **intra-modal asymmetric text retrieval task**. This design addresses the limitation of CLIP-based **vision-to-text matching** methods, which overlooking fine-grained attribute bindings of objects due to the “*bag-of-words (BoW)*” behavior.
* **Novel Benchmark Contribution (Reviewer v764,  uodS, and AMGd).** Prior benchmarks are ill-suited for this task in that the queries are **simple category labels**. To address this critical gap, we construct Fine-OVS, a novel benchmark with queries annotated across **8 fine-grained attribute types**.
* **Extensive Convincing Experiments (Reviewer AMGd).** Extensive quantitative and qualitative experiments conducted on the Fine-OVS demonstrate  that our FineSplat framework significantly outperforms existing state-of-the-art methods.

We've revised our paper according to the reviewers' suggestion (**highlighted in blue in the uploaded revision pdf**). Below summarize the major updates we've made:
* **More Quantitative Comparisons.**
  1) *Quantitative Comparisons on the LERF* **(Reviewer v764 and uodS)**. FineSplat not only retains the fundamental coarse-grained query capabilities but also successfully models more complex fine-grained feature fields. The experimental results are shown in Table 9 **(Appendix D.2)**.
* **More Ablation Study.**
  1) *Ablation Study of the Text Encoder* **(Reviewer uodS and AMGd)**. To demonstrate the effectiveness of the text encoder used in the FGFFM, we conduct additional experiments using different combinations of text encoders, and the results are shown in Table 11 and Table 12 **(Appendix E.4)**.
  2) *Computational Cost Analysis* **(Reviewer v764)**. We conduct a full computational cost analysis for the entire pipeline on a single 24G RTX-4090 GPU, as shown in the Table 13 **(Appendix E.6)**.
  3) *The Generalization Ability of FineSplat* **(Reviewer uodS)**. We conduct a more targeted generalization study to further validate the generalization capability of FineSplat. The results are shown in the Table 16 and Table 17 **(Appendix E.7)**.
* **More Visualizations.**
  1) *Multi-view Consistency Study* **(Reviewer uodS)**. To better demonstrate that FineSplat produces consistent captions across multiple views, we visualized the feature maps from different views, and the results are shown in the Figure 10 **(Appendix E.2)**.
  2) *Fine-grained Attribute Awareness of FineSplat* **(Reviewer uodS)**. We select examples that satisfy same category but differing in only one fine-grained attribute and examine FineSplat’s localization results for these test queries,  as shown in Figure 12 **(Appendix E.5)**.

We believe we have thoroughly addressed all of the reviewers' concerns. We hope that our work can significantly advance research in fine-grained scene understanding tasks. Details on these updates and point-by-point responses to individual concerns can be found in the response to each reviewer below.

We hope this summary assists in your final assessment. Thank you for your time and consideration.

Best regards,

Authors

---

### Meta-Review · Area_Chair_HdzZ · 2026-01-04

**Summary:**

The reviewers’ concerns that informed the decision can be summarized as follows:
1. The proposed method relies on a multi-stage, computationally heavy pipeline involving SAM, DAM, and large MLLMs for caption generation, followed by scene-specific training components. Reviewers questioned whether this constitutes a scalable or principled modeling advance.

2. FineSplat requires per-scene training (including a scene-specific autoencoder), limiting its applicability as a general-purpose model. This raises concerns about whether the method advances open-vocabulary 3D understanding beyond narrow settings.

3. The newly introduced Fine-OVS benchmark is small (8 scenes) and constructed by the authors. Strong performance is demonstrated primarily on this custom benchmark, while results on standard datasets (e.g., LERF) are weaker, suggesting over-specialization.

4. Reviewers were concerned that improvements in fine-grained queries come at the expense of performance on standard coarse-grained queries, calling into question whether the method improves general scene understanding or merely introduces a niche capability.

These concerns go beyond presentation or missing experiments and relate to the fundamental scope and impact of the work.

**Reviewer Concerns:**

Concerns partially addressed by the rebuttal

1. The authors provided a detailed runtime and memory analysis, which improves clarity but does not fundamentally change concerns about scalability or practicality.

2. New ablations (e.g., text encoder choices, captioning components) and visual results help clarify the method’s behavior but largely reinforce that performance depends on a complex, carefully tuned pipeline.

3. The authors demonstrated that coarse-grained performance can be recovered with additional architectural adjustments, but this further increases system complexity and weakens the claim of a unified solution.

Concerns still outstanding

1. The rebuttal does not convincingly address whether the method can scale beyond small, per-scene settings or be applied efficiently to larger, real-world datasets.

2. The Fine-OVS benchmark remains small and author-curated, with no validation on larger or more diverse datasets that would support strong general claims.

**Reviewer Scores:**

Reviewer v764 (initial: 4  marginally below threshold):
Likely unchanged. While questions were answered, the reviewer’s core reservations about complexity, scalability, and niche applicability persist.

Reviewer uodS (initial: 4 marginally below threshold):
Likely unchanged. Additional ablations address some questions, but concerns about multi-view consistency and true semantic understanding are not fundamentally resolved.

Reviewer AMGd (initial: 8 accept):
Likely unchanged, but unlikely to outweigh the fundamental concerns raised by more critical reviewers.

---

### Decision · Program_Chairs · 2026-01-26

Reject